# Development of Olive Oil Containing Phytosomal Nanocomplex for Improving Skin Delivery of Quercetin: Formulation Design Optimization, In Vitro and Ex Vivo Appraisals

**DOI:** 10.3390/pharmaceutics15041124

**Published:** 2023-03-31

**Authors:** Omnia M. Hendawy, Mohammad M. Al-Sanea, Rehab Mohammed Elbargisy, Hidayat Ur Rahman, Hesham A. M. Gomaa, Ahmed A. B. Mohamed, Mohamed F. Ibrahim, Abdulsalam M. Kassem, Mohammed Elmowafy

**Affiliations:** 1Department of Pharmacology, College of Pharmacy, Jouf University, Sakaka 72341, Saudi Arabia; 2Department of Pharmaceutical Chemistry, College of Pharmacy, Jouf University, Sakaka 72341, Saudi Arabia; 3Department of Pharmaceutics, College of Pharmacy, Jouf University, Sakaka 72341, Saudi Arabia; 4Department of Clinical Pharmacy, College of Pharmacy, Jouf University, Sakaka 72341, Saudi Arabia; 5Department of Medicinal Chemistry, Faculty of Pharmacy, Mansoura University, Mansoura 35516, Egypt; 6Department of Pharmaceutics and Pharmaceutical Technology, Faculty of Pharmacy (Boys), Al-Azhar University, Cairo 11651, Egypt

**Keywords:** phytosomes, skin delivery, quercetin, olive oil

## Abstract

The objective of the current work was to fabricate, optimize and assess olive oil/phytosomal nanocarriers to improve quercetin skin delivery. Olive oil/phytosomal nanocarriers, prepared by a solvent evaporation/anti-solvent precipitation technique, were optimized using a Box–Behnken design, and the optimized formulation was appraised for in vitro physicochemical characteristics and stability. The optimized formulation was assessed for skin permeation and histological alterations. The optimized formulation (with an olive oil/PC ratio of 0.166, a QC/PC ratio of 1.95 and a surfactant concentration of 1.6%), and with a particle diameter of 206.7 nm, a zeta potential of −26.3 and an encapsulation efficiency of 85.3%, was selected using a Box–Behnken design. The optimized formulation showed better stability at ambient temperature when compared to refrigerating temperature (4 °C). The optimized formulation showed significantly higher skin permeation of quercetin when compared to an olive-oil/surfactant-free formulation and the control (~1.3-fold and 1.9-fold, respectively). It also showed alteration to skin barriers without remarkable toxicity aspects. Conclusively, this study demonstrated the use of olive oil/phytosomal nanocarriers as potential carriers for quercetin—a natural bioactive agent—to improve its skin delivery.

## 1. Introduction

Herbal medicines have received attention worldwide in the past few years. Quercetin (QC; 3,3′,4′,5,7-pentahydroxyflavone; Figure 1), the major constituent of several natural plants [1], is considered one of the most powerful natural antioxidants [2] due to its free radical scavenging, prevention of lipid peroxidation, metal ion chelation and modulation of cell antioxidant responses [3]. It also exerts anti-inflammatory [4], anticancer, antibacterial [5] and antiviral effects [6]. Taking into account the action of QC on the skin, it has been reported that its topical application decreases the cutaneous oxidative damage induced by free radical overproduction [7,8]. In particular, QC prohibited the skin’s inflammatory reaction, histological alterations and the expression of matrix metalloproteinases generated by solar ultraviolet radiation [7,8]. Furthermore, quercetin was found to reduce human melanoma cell growth [9], produce anti-inflammatory action on human keratinocytes [10] and activated neutrophils [11]. Unfortunately, all these benefits cannot be effectively utilized without designing suitable topical drug-delivery systems, as QC suffers from poor skin penetration [11,12,13], which precludes its bioavailability after application to viable skin [8,12,14]. This behavior is mainly related to its limited solubility in water and concurrently in oils [12,15]. Several approaches have been developed, such as the use of permeation enhancers [16,17], the synthesis of prodrugs [11], the fabrication of microemulsions [8,12], the development of liposomes [14,18] and the development of polymeric [19] or lipid nanoparticles [13,15]. In the current study, a different strategy is outlined which involves utilizing olive oil containing phytosomal complexes to improve the skin permeation of QC.

Phytosomes are promising phospholipid-based nanocarriers loaded with natural bioactive agents [20,21]. According to another concept, such systems involve complexation between phospholipids and a guest polyphenolic active agent. Therefore, these systems are considered phospholipid/drug-biocompatible supramolecular complexes that self-assemble into vesicular structures upon the addition of aqueous media [22]. Guest polyphenolic active agents are constructively bonded by electrostatic and H-bonds with the phospholipid polar head groups (phosphate and ammonium groups) [23]. Phytosomal complexes present high drug-loading capacities and are easily stored in solid/lyophilized forms ready for reconstitution prior to use [21]. Phospholipids are well-known for their safe and effective skin delivery potential [24]. On the other hand, olive oil is a natural vegetable oil rich in fatty acids, particularly oleic acid and linoleic acid [25]. These fatty acids have the ability to enhance skin permeation through alteration of the fluidity of skin lipid structures [26].

To the best of our knowledge, none of the previous studies in the literature have systematically studied the simultaneous influence of olive oil/phospholipid ratio, quercetin/phospholipid ratio and the concentration of surfactant on the physicochemical behavior of developed phytosomes in order to improve skin delivery. Therefore, it appears reasonably well justified to develop a phytosomal complex. The first step was to optimize the formulation constituents through studying the influence of olive oil/phospholipid ratio, quercetin/phospholipid ratio and the concentration of surfactant on particle size, zeta potential and EE%. The optimized formula was further investigated for drug excipient interaction, stability and in vitro release. The second step was to evaluate the optimized formula by studying ex vivo skin permeation and skin histological changes. 

## 2. Materials and Methods

### 2.1. Materials

Quercetin (QC) and Pluronic F127 (Poloxamer 407; PF-127) were purchased from Sigma-Aldrich (Karnataka, India). Olive oil and Tween 80 were purchased from Loba Chemie (Mumbai, India). Phospholipid (Phospholipon^®^ 90 G; PC) was kindly gifted from Lipoid (Steinhausen, Switzerland). Acetone (purity > 99%), ethanol (95%) and n-hexane (purity ≥ 97%) were purchased from Sigma-Aldrich (Steinheim, Germany). All other chemicals and solvents were in analytical grades.

### 2.2. Fabrication of Olive Oil/Phytosomal Complexes 

Olive oil/phytosomal complexes loaded with QC were prepared by solvent evaporation/anti-solvent precipitation technique [27]. Firstly, phospholipid, QC and olive oil were dissolved in 20 mL acetone with continuous stirring until the complete evaporation of acetone took place. Then, the residual was dissolved in 10 mL of n-hexane using continuous stirring in the fume hood. Complete removal of n-hexane was verified by weighing the mixture prior to solvent addition and after its evaporation [28]. Phytosomal complexes (F2–F10) were developed by the addition of aqueous solution of PF-127 (in different concentrations; 0.5%, 1% and 2% *w*/*v*) to hydrate the thin film followed by probe sonication (Crest Ultrasonic Corp., Ewing Township, NJ, USA) for 2 min [29]. The formulations were further dispersed by vortexing (Dragonlab cortex, Beijing, China) to obtain a homogenous dispersion. 

### 2.3. Experimental Design

Phytosomal complexes were designed and optimized by Box–Behnken design using the Design Expert 13 software, which produced 17 experiments including 12 factorial and 5 center points. Depending upon initial results, the ratios and percentages of constituents were selected to achieve the desired particle diameter, surface charge and entrapment efficiency %. The three factors, designated as an olive oil/PC ratio-A (1:10–1:2), a QC/PC ratio-B (2:1–1:2) and a concentration of surfactant-C (0.5–2%), were predicted to affect the process of Phytosomal complexes’ formulation (Table 1). The responses included the particle diameter (Y1), surface charge (Y2) and entrapment efficiency % (Y3) of the formulations. The optimization of independent variables was designed to decrease the particles’ diameter (Y1), and increase both surface charge (Y2) and encapsulation efficiency (Y3). The compositions of the all phytosomal complexes are outlined in Table 2 with 1% of QC in the final batches.

### 2.4. Particle Diameter and Surface Charge 

The dynamic light scattering technique was used to measure particle diameter and surface charge by Zetasizer analyser (Malvern Zetasizer Nano ZS90, Worcestershire, UK) at room temperature and at 90° as scattering angel [30]. Phytosomal complexes were diluted 10-fold with double distilled water and vortexed. In the case of particle diameter measurements, disposable polystyrene cuvettes were used, while disposable folding capillary cuvettes were used for the surface charge measurements. All samples were measured in triplicate in three runs after the device completed the stabilization.

### 2.5. EE%

The encapsulation efficiency of phytosomal complexes was determined by an indirect method reported by El-Refaie et al. [31] after the separation of un-entrapped QC. Briefly, dispersions of phytosomal complexes were centrifuged at 15,000 rpm for 15 min at 4 °C. The un-encapsulated QC was quantitatively determined in supernatant by the HPLC system. Briefly, the collected supernatants were filtered through a 0.20 μm syringe filter and the filtrates were injected into the HPLC system (Thermo Scientific Dionex UltiMate 3000 UHPLC+, Carlsbad, CA, USA) equipped with C18 column (150 mm, 4.6 mm and 5 μm particles size), and then the samples were detected by DAD-3000 diode array detector at wavelength of 210 nm. The flow rate was 0.7 mL/min. The mobile phase was a mixture of phosphoric acid acidified water (pH~3) and acetonitrile: (60:40 *v*/*v*) in isocratic elution mode. The method was validated by the determination of linearity, precision, accuracy, limit of detection (LOD) and limit of quantification (LOQ). Linearity was determined by adding a mobile phase to various concentrations (0.02 µg/mL to 80 µg/mL). Four QC concentrations (0.1, 0.5, 5 and 20 µg/mL) were utilized to determine precision and accuracy. LOD and LOQ were assessed by the standard deviation of intercept (SDb) and slope (a) of calibration curve (please see Appendix A). EE% was calculated using the following formula:$$\mathrm{EE}\%=\frac{\mathrm{Total}\;\mathrm{amount}\;\mathrm{of}\;\mathrm{QC}\;\mathrm{added}-\;\mathrm{amount}\;\mathrm{of}\;\mathrm{QC}\;\mathrm{in}\;\mathrm{supernatant}}{\mathrm{Total}\;\mathrm{amount}\;\mathrm{of}\;\mathrm{QC}\;\mathrm{added}}\times 100$$

### 2.6. Investigation of QC/Excipients Interaction 

The possibility of interaction between the other components of the formulation was assessed by Fourier transform infrared spectroscopy (ATR-FTIR; Thermo Scientific) and differential scanning calorimetry (DSC, DSC3, Mettler Toledo, Columbus, OH, USA). In the case of ATR-FTIR, the investigation of interaction depends on the recording of the characteristic peak wavelength of the function groups. The QC, phospholipid, PF-127, olive oil and optimized formulation were scanned in the range between 4000 and 400 cm^−1^ in transmission mode. In the case of DSC, the interaction depends upon the change in thermal peak of transition. The QC, phospholipid and optimized formulation were scanned at temperatures ranging from 30 °C to 350 °C with a heating rate of 20 °C/min.

### 2.7. Stability Studies

The optimized formulation was assessed for stability during its storage for three months at ambient temperature (25 ± 2 °C) and refrigerating temperature (4 °C). The stability of the optimized formulation (as nanodispersion) was determined according to the changes in particle diameter, polydispersity index (PI), EE% and zeta potential. In order to compare it with conventional phytosomal complexes, one formulation (F0) was subjected to the same study. F0 (conventional QC/phospholipid phytosomal complex) was prepared using the same procedure as the optimized formulation without the incorporation of olive oil or PF-127.

### 2.8. In Vitro QC Release 

The in vitro QC release studies of the optimized formulation, F0 and QC dispersion (prepared by dispersion of QC in 0.5% methylcellulose) were carried out using a vertical Franz-diffusion cell apparatus with an autosampler (Logan DHC-6T Dry Heat Transdermal System) and using temperature control. The donor compartment composed of 2 mL of the formulations which were placed onto the cellulose acetate membrane (M. wt. cut off: 12–14 KDa) were fixed to the terminal side of the apparatus sets. The receptor compartment was composed of 12 mL of PBS/ethyl alcohol (70:30 *v*/*v*; pH = 7.4) to maintain the sink conditions of QC and kept at 37 ± 1 °C. Aliquots from the receptor compartment were withdrawn at time intervals (0.5, 1, 2, 3, 4, 6 and 8 h), and QC was quantified using the same method mentioned in Section 2.5 EE%. The average values of 3 measures (±SD) were recorded. Equivalent volumes to the withdrawn ones were substituted with fresh medium to maintain the sink conditions. In order to investigate the appropriate release mechanism of QC, the data were fitted to different kinetic models (zero order model, first order model, Higuchi diffusion and Hixson–Crowell model kinetic equations) to investigate the best-fitting model for the release data.

### 2.9. Ex Vivo Skin Permeation

Ex vivo skin permeation has been carried out for the optimized formulation and F0 in comparison with QC dispersion as a control. All formulations were used in a dose equivalent to (20 mg). The experiment was conducted on an isolated dorsal side of Albino rabbits (male rabbits with an average weight of 1.75–2 kg) after hair removal. The animal studies were performed according to the ethical procedures and policies of Jouf University (Code; 06-05-42) using Franz-type diffusion cells (Logan DHC-6T Dry Heat Transdermal System) fortified with a temperature regulator (kept at 37 ± 1 °C), using autosampling, and using a self-reservoir replacement technique. The isolated skin specimens, devoid of hairs, were washed with Ringer solution (composed of 0.12% sodium chloride, 0.062% sodium lactate, 0.006% potassium chloride and 0.004% calcium chloride) following which they were mounted onto the sets where the stratum corneum encountered the formulation. The receptor compartment consisted of 12 mL of PBS/ethyl alcohol (70:30 *v*/*v*; pH = 7.4) facing the endoderm. At predetermined time points (30, 60, 120, 240, 360 min), aliquots of 200 μL were withdrawn to be analyzed by UPLC and replaced by equal volumes of the reservoir solutions. At the end of the experiment, the skin specimens were washed 5 times with PBS/ethyl alcohol (70:30 *v*/*v*; pH = 7.4) and dried. The QC deposited in the skin was quantified by the previously mention UPLC method after extraction. The extraction was carried out by cutting the skin specimens into small pieces and then performing a homogenization (Ultra-Turrax homogenizer, IKA, Staufen, Germany) in PBS/ethyl alcohol at 10,000 rpm for 15 min. The resulting suspensions were centrifuged for 15 min at 5000× *g* and the supernatants were filtered to determine QC quantity [32]. 

### 2.10. Skin Compliance 

In this section, we assessed the safety of optimized formulation and F0, in comparison with the untreated group, by both visual observation and skin histological changes. Briefly, the investigated formulations were topically applied to the shaved skin areas on the dorsal side of the rabbit (*n* = 6) for 10 days. On the third, seventh and tenth day, the animals were assessed for any manifestation of skin redness (the score ranged from 0 to 4 depending upon the severity of the redness). After that, the animals were sacrificed and the areas of investigation were removed for the histopathological investigation by standard Hematoxylin and Eosin staining. 

### 2.11. Statistical Analysis

All results are displayed as means ± standard deviations (SD). The statistical analysis was carried out by a one-way ANOVA and means were compared using Tukey’s multiple comparison testing using GraphPad Prism v.5. When *p* values were less than 0.05, it was considered a significant difference.

## 3. Results and Discussion

### 3.1. Rationale and Formulation of QT Loaded Olive Oil/Phytosomal Nanocarries

Among phospholipid-based systems, phytosomes were suggested to improve QC skin delivery. Indeed, phytosomes have been utilized by many researchers to improve the limited oral bioavailability of phytochemicals [29,33,34]. However, to the best of our knowledge, utilizing the system for improving skin delivery has only been performed in a few studies. Phytosomes possess the promising credentials to be efficient carriers for cutaneous delivery as carrier materials have the most important role in order to perform the desirable effect. In particular, phospholipids have similar structures to the skin leading to their fusion to skin lipids and then to the diffusion of drug molecules across skin layers [35]. Additionally, nanometric systems can easily diffuse through the skin. Herein, we incorporated olive oil in phytosomal nanocarrier formulations as olive oil was reported to improve skin penetration [25]. We studied the influence of olive oil/PC ratio, QC/PC ratio and concentration of surfactant on particle size, zeta potential and EE%. 

The formation of the QC-loaded olive oil/phytosomal nanocarrier is the most essential procedure in the fabrication of the system. In this work, we used the solvent evaporation technique to develop QC-loaded olive oil/phytosomal nanocarriers [27]. In our study, we used acetone as the reaction solvent because it was reported to organize and not disrupt the proton exchange process. We also used n-hexane antisolvents to separate phytosomal complexes by precipitating them out from the organic solvent [36,37]. When the phytochemical (QC) in a defined quantity is dissolved in acetone with phospholipid, the weak interaction between the polar heads of the phospholipid with QC forms the phytosomal complex. By continuing, more complexes would be formed. 

### 3.2. Box–Behnken Design Analysis and Optimization

In this design, 17 runs of experiments were developed in triplicate. The findings of particle diameter (Y1), surface charge (Y2) and EE% (Y3) were outlined Table 2. A randomized response surface study was utilized to develop the whole set of formulations and remove any chance of biases. Expressions produced by the non-linear quadratic model using Box–Behnken design was found to be the best model to interpret the effect of independent variables (including olive oil/PC ratio, QC/PC ratio, and concentration of surfactant) on the experimental results. Insignificant lack of fit F-values (*p* > 0.05) as well as small F-values in all responses augmented appropriate fitting to quadratic model (Table 3).

#### 3.2.1. Effect of Independent Variables on Particle Size 

The particle size of the nanoparticulate system is an important parameter for the skin delivery of carried drug molecules. Commonly, the smaller the particles’ diameter, the higher the adhesion and occlusion exhibited by the system [38]. It is clear that the particle diameter of the selected 17 runs varied between 288 and 208 nm (Table 4), depending on the level of independent variables and interactions between each other. Run 14 exhibited the highest (287.9 nm), while Run 10 displayed the smallest particle diameter (208.5 nm). 

To obtain a better consideration of the three effects of the independent variables and their interactions on the responses, three-dimensional (3D) response surface plots for the particle size were plotted based on the model equation below using the Expert Design 13.0.12.0 software (Figure 2). The analysis of variance (ANOVA) results (Appendix A) show the significant effect of all the studied factors individually, which are olive oil/PC ratio-A, QC/PC ratio-B and the concentration of surfactant-C on the particle diameters of the batches, while only the interaction between olive oil/PC ratio-A and QC/PC ratio-B was observed to have a significant effect on the particle diameter.

The polynomial equation of the particle size in quadratic expression is given below:Y_1_ = +228.90 + 10.70A − 13.07B − 14.65C + 22.90AB − 1.50AC − 3.85BC + 13.25A^2^ + 13.35B^2^ − 3.80C^2^

The formula designates that the diameter of phytosomal complex increases with the olive oil/PC ratio (A) while the QC/PC ratio (B) and the concentration of surfactant (C) had an inverse relation with the diameter of phytosomal complex particles. The coefficient of variance of center points (Runs 2, 4,9,12, and 16) was of 2.69% indicating better reproducibility. In Figure 2, it is observed that at higher ratios of the olive oil/PC, there was an increase in the particle diameter. Increment in the particle diameter upon the increase in olive oil/PC ratio might be attributed to the inability of phospholipid to emulsify high proportions of olive oil, especially, at low concentrations of surfactant (0.5%). This was clear when increasing the olive oil/PC ratio from 0.1 to 0.5, which resulted in the increase in particle diameter from 237.7 to 268.9 nm at low concentrations of surfactant (0.5%). Although phospholipid is a well-thought-out stabilizer, it is not capable of stabilizing the system effectively [39]. Hence, the addition of olive oil in high concentrations without the effective concentration of surfactant will lead to an inappropriate decrease in tension at the interface and promote agglomeration. Similar findings were obtained by Manca and coworkers, who studied the incorporation of argan oil in phospholipids to develop a liposome-like formulation [40]. The authors ascribed this behavior to the oil effect on bilayer assembling and structure, which preferred the development of vesicles with a higher curvature radius. Song and coworkers studied flaxseed oil incorporation into phospholipids, focusing on structural changes in bilayers [41]. The authors attributed the increase in particle size upon excessive addition of flaxseed oil to the destruction of the bilayer structure. On the other hand, a low QC/PC ratio resulted in an increase in particle size. This behavior was due to high concentrations of phospholipids that led to an increase in dispersion viscosity, resulting in higher surface tension and, hence, larger particle sizes [42]. In addition, and according to Stock’s law, high viscous dispersion resists particle collision and may lessen the chance of it breaking down into smaller particles [43]. Regarding the surfactant concentration, increasing the concentration decreased the particle diameter. As mentioned above, the addition of a surface active agent to the phytosomal complex can lead to effective stabilization of the system by covering the phospholipid surface efficiently, leading to the reduction in the interfacial tension between solvent phases [44]. In particular, PF-127 was reported to be strongly adsorbed on the surface of nanosystems, modifying their surface properties. This adsorption of PF-127 could reduce the Gibbs free energy and alter the hydrophilicity of the substance, which improves steric repulsion and, hence, the stability of the nanosystems [45].

#### 3.2.2. Effect of Independent Variables on Surface Charge

Surface charge is commonly presented by zeta potential [46]. Zeta potential is another crucial factor during the design of nanoparticulate systems. Commonly, the greater the absolute potential value, the higher the degree of repulsive force between particles and the higher system stability. It is obvious that the values of zeta potential of the selected 17 runs varied between −33 and −21.4 mV (Table 4), based upon the level of independent variables and interactions between each other. Run 14 exhibited the highest zeta potential value (−33 mV), while the Run 17 showed the lowest zeta potential value (−21.4 mV) towards negative direction. It was reported that the nanoparticulate system could be efficiently stabilized if zeta potential is of −30 mV when the system is electrostatically stabilized. If the system is sterically stabilized a value of −20 mV is sufficient [47]. Figure 3 displays 3D response surface plots for the zeta potential, representing the influence of independent variables. The ANOVA results (Appendix A) show that olive oil/PC and QC/PC ratios significantly influenced the zeta potential value of the formulations.

The polynomial equation of zeta potential in the quadratic expression is given below:Y_2_ = −27.96 + 1.22A + 1.29B − 0.4625C − 2.27AB − 1.93AC − 0.1500BC + 1.46A^2^ − 0.6200B^2^ + 1.38C^2^

The formula describes the effect of different individual variables on the zeta potential value of the phytosomal complex. In Figure 3, it is observed that at higher ratios of olive oil/PC and QC/PC, there were decreases in the value of zeta potential. In general, the negativity of zeta potential is gained by the presence of the negative phosphate group in phospholipid, which is directed to the outside layer of the phytosomal complex [48]. Therefore, the degree of negativity of the zeta potential value is directly related to the density of phospholipids on the surface of the phytosomal complex [49]. As olive oil/PC ratio increased, the negativity of the zeta value decreased due to the presence of triglycerides and fatty acids [28] in olive oil. The high negative value was obtained by the formulation containing the least concentration of olive oil (Run 14). This result is in good agreement with Komeil et al. [28]. On the other hand, Manca and coworkers found opposite results upon combining argan oil with phospholipid [40]. On the other hand, high QC/PC ratio produced less negative value of zeta value. The phytosomal complex is basically formed by formation of hydrogen bonds between the polar parts of phospholipid (such as phosphate and ammonium groups) and the polar groups of the polyphenol phytochemicals [50]. However, this interaction is supposed to be the reason for the partial fading of the negativity on the surface of the phytocomplex in the case of incorporation of high QC amounts. This finding was in good accordance with the previous study by Angelico et al. [51] who studied the development of silybin–phospholipid phytocomplexes. The optimum ratio between the phytochemical and phospholipid was reported to be 1:1 or 1:2 [52]. The concentration of surfactant had insignificant effects (*p* > 0.05) on the zeta potential value of the developed formulations.

#### 3.2.3. Effect of Independent Variables on EE%

EE% is an important parameter for the evaluation of nanoparticulate systems. In general, phytosomal complexes are expected to have higher drug loading properties when compared to other vesicular systems due to the formation of a H-bond between the polyphenol and the polar heads of phospholipids leading to the association of polyphenol with both the interior and exterior membrane of the phytosomes [53]. EE% of the selected 17 runs varied between 72.9 and 92.1% (Table 4) based upon the level of independent variables and interactions between each other. Run 13 exhibited the highest EE%, while Run 17 showed the lowest EE% among all investigated batches. Figure 4 displays the 3D response surface plots for the EE% representing the influence of independent variables. The ANOVA results (Appendix A) show that olive oil/PC ratio, QC/PC ratio and concentration of surfactant significantly influenced EE% of the formulations.

Regarding EE, the polynomial formula is:Y_3_ = +84.82 − 1.46A + 1.64B + 3.85C + 0.0750AB + 2.80AC + 0.4500BC − 4.95 A^2^ − 0.0975B^2^ + 1.23C^2^

The equation defines the effect of different individual variables on the EE% of the phytosomal complex. In Figure 4, it is observed that by increasing the ratio of the olive oil/PC, there was initially an increase followed by a decrease in the EE%. This behavior might be attributed to the improvement in drug distribution in the phospholipid bilayers by using low concentrations of olive oil. On the other hand, using a high olive oil/PC ratio is expected to increase the fluidity of bilayers [41] and interfere in the phospholipid/phytochemical interaction leading to a leakage of the active drug and a decreased EE. Regarding QC/PC ratio, it was noticed that the higher the ratio, the higher the EE%. This reflects that there is enough space in the phospholipid bilayer to accommodate for QC efficiently in its maximum level. It is suggested that while the phospholipid/QC complex is dissolved in the organic solvent, the hydrophilic head of the phospholipid is adapted to the hydrophilic group of the QC and the hydrophobic tail of phospholipid is adapted to the hydrophobic media to configure to the appropriate directional orientation [54]. In addition, QC has a good capability for interaction with phospholipid as the former possesses five hydrogen bond donors and seven hydrogen bond acceptors which are more than other polyphenols. Hence, the chance of QC forming H-bond complexations with phospholipid was anticipated to be high leading to high incorporation of QC in the phytosomal complex [55]. The concentration of the surfactant had a significant (*p* < 0.05) effect on the EE% of the developed formulations. The higher the PF-127 concentration, the higher the encapsulation efficiency percentage obtained. As mentioned above, PF-127 was reported to be strongly adsorbed on the surface of nanosystems, modifying their surface properties and leading to decreasing the Gibbs free energy [45]. It was reported that higher concentration of surfactant could decrease EE% due to participating in particle size reduction [56] and drug partitioning into aqueous phase [57]. Additionally, it was reported that using Pluronics in high concentrations (above the critical micelle concentration (0.26–0.8 wt%; [58])) led to transformation into flat discs at temperatures above transition temperature, but stayed spherical shaped at temperatures below transition temperature [59]. In our case, olive oil was incorporated, so the system was a binary system and needed the incorporation of surfactant to reduce the interfacial tension and form a stable system as phospholipid is not sufficient as a surface active agent in such cases. This might partially explain the high EE% upon using high surfactant concentrations. 

#### 3.2.4. Optimization and Point Prediction

The optimization of the independent variables and then point prediction was performed via the desirability method. The numerical optimization was accomplished through identifying the optimal levels of each independent variable (designated as A, B and C), which achieves the desired outcome in the dependent variables (minimizing Y1, maximizing the value of Y2 and maximizing Y3). Five independent lots of the optimized formula were developed as recommended by the utilized software. Responses, in terms of particle diameter (nm), zeta potential (mV) and EE%, of predicted and actual values are outlined in Table 5 and Appendix A. It is clear from the presented records that the predicted values of the responses were in good accordance with the actual measurements elucidating the dependability on the optimization process for the preparation of QC loaded olive oil/phytosomal complex [60]. Hence, the optimized composition was found to be an olive oil/PC ratio of 0.166, a QC/PC ratio of 1.95 and a surfactant concentration of 1.6%. This composition showed a reasonable particle diameter, zeta potential value and satisfying EE% (Table 5).

### 3.3. FT-IR

An interaction between QC and other formulation components was investigated by FT-IR spectroscopy. Figure 5 represents the spectra of QC, phospholipid, olive oil, PF-127 and optimized formulation. The FT-IR spectrum of pure QC showed the main signals at 3361 cm^−1^ (corresponding to free hydroxyl bond vibration), 1627 cm^−1^ (corresponding to carbonyl group stretching), 1597 cm^−1^ (corresponding to C–C stretching of phenyl ring), at 1242 cm^−1^ (corresponding to phenolic –OH bending), 1157 cm^−1^ (revealing C–O–C vibration) and at 995 cm^−1^ (corresponding to aromatic C–H) [61]. The phospholipid presented a characteristic broad signal at 3309 cm^−1^ (corresponding to –OH vibrational stretching [62]), at 2916 and 2854 cm^−1^ (corresponding to C–H stretching in long chain fatty acid), 1735 cm^−1^ (corresponding to C=O stretching of fatty acid ester), 1234 cm^−1^ (corresponding to –P=O stretching) and 962 cm^−1^ (corresponding to –N^+^(CH_2_)_3_ stretching). The olive oil spectrum displayed sharp signals at 2921 cm^−1^ and 2853 cm^−1^ (corresponding to C–H stretching vibrations) and 1741 cm^−1^ (C=O stretching of fatty acid ester) [63]. In the spectrum of the optimized formulation (composed of an olive oil/PC ratio of 0.166, a QC/PC ratio of 1.95 and a surfactant concentration of 1.6%), the fainting and shifting of some peaks were observed. In particular, the broadening and shifting of the representative phenolic hydroxyl bond vibration of QC from 3361 cm^−1^ to 3294 cm^−1^ could be potentially ascribed to the formation of hydrogen bonding [21]. Additionally, the shifting of –P=O stretching peak (1234 cm^−1^) of phospholipid was detected in the spectrum of the optimized formulation (1126 cm^−1^). This finding pointed to some interactions between QC and phospholipids. Conclusively, the FT-IR technique proved the interaction between QC and phospholipid molecules in phytosomal complexes.

### 3.4. DSC

DSC analyses of QC and lyophilized-optimized formulation were carried out in order to investigate the degree of crystallinity of QC after the inclusion of olive oil containing phytosomal complexes (Figure 6). However, liquid lipids, such as olive oil, could not be recorded via the distinct temperatures and analytical settings [64]. The raw QC displayed one principal sharp endothermic at 316°, approving its crystallinity [25]. The phospholipid thermogram showed three peaks at 153.3 °C, 165.3 °C and 238.5 °C designating the transition from a gel-state to a liquid crystalline-state and the melting of carbon-hydrogen chains into phospholipids [65]. Interestingly, the optimized formulation (composed of an olive oil/PC ratio of 0.166, a QC/PC ratio of 1.95 and a surfactant concentration of 1.6%) showed faint broad peaks at 130 °C, 137 °C and 284 °C. Firstly, the disappearance of QCs sharp peak and the appearance of a new faint broad peak at 284 °C in the thermogram proved the existence of QCs in amorphous state which could enhance QC solubility. It was reported by many researchers that phospholipid polar groups can interact with polar groups of polyphenolic’s natural compounds by hydrogen bond or hydrophobic interaction and not by chemical or hybrid bonds [66,67]. Secondly, the shifting of phospholipid endothermic peaks into lower temperatures with lower intensities is suggested to be due to incorporation of olive oil which decreased the consistency of the system [68], which could in turn increase the amorphousness state of the phospholipid. However, it was reported that the interaction between the polyphenol and the phospholipid polar groups promoted the free rotation of the carbon-hydrogen chain of phospholipid, leading to a decrease in phospholipid aliphatic hydrocarbon chains. This could result in the disappearance of the endothermic peaks of phospholipids and the decrease of the phase transition temperature [65].

### 3.5. Stability Studies

The stability of the optimized formulation and F0 was evaluated in terms of particle diameter (nm), PDI, zeta potential (mV) and EE%. The stability of both formulations was examined at ambient and refrigerating temperatures for 3 months (Table 6). The results confirmed that phytosomal complexes, in general, are better stored at room temperature than at refrigeration temperature. The aggregation of phytosomal complexes (the optimized formulation and F0) was observed at 4 °C with a significant growth (*p* < 0.05) in particle size (243.8 ± 16.4 and 295.7 ± 18.7 nm, respectively), as well as increase in PI (0.30 ± 0.04 and 0.36 ± 0.05, respectively) after 3 months. However, phospholipid-based colloidal systems have a higher tendency to associate to form larger particles when stored at refrigeration temperature than ambient temperature [69]. On the other hand, there was an insignificant decrease (*p* > 0.05) in EE% during the storage period, which might be attributed to the complete and tight complex formation of QC with phospholipid, which prohibited QCs escape during the storage [70].

Regarding the optimized formulation, it was observed that particle size and PI wereincreasing less with time than those of F0. Efficient adsorption of PF-127 on the particles’ surfaces led to a decrease in the Gibbs free energy and an increase in steric repulsion, resulting in resisting particles aggregation and an enhanced stability [45].

### 3.6. In Vitro QC Release Studies

During the in vitro QC release from the optimized formulation, F0 and QC dispersion was appraised over a period of 8 h (Figure 7). It was obvious that the release percentage of QC from the dispersion (33.3 ± 2.7) was significantly (*p* < 0.05) lower than that of both phytosomal complexes (optimized formulation and F0) after 8 h. Both phytosomal complexes showed a biphasic release pattern: there was an initial burst release within the first 2 h (around 37.4% and 30.5% for the optimized formulation and F0, respectively) followed by a controlled release manner during the last 6 h. The initial fast release pattern might be attributed to the free (in non-complex form) QC or loosely attached QC molecules to phytocomplex surfaces while the sustained release pattern was attributed to the tight complexation of the QC with phospholipid [71] leading to the gradual QC dissociation from the phytocomplex and its consequent diffusion to the receptor compartment through the dialyzing membrane [29]. Similar findings were obtained by Alhakamy et al. [72], showing the release pattern of thymoquinone from phytosomal complexes. On the other hand, the results presented that the QC release reached 69.3% in the case of the optimized formulation, while the QC release reached 51.6% from F0 after 8 h. The significantly higher levels of QC released from the optimized formulation (*p* < 0.05), when compared to F0, might be attributed to the incorporation of olive oil in the complex which increased the fluidity of the system leading to the easier escape of the encapsulated QC. As shown in Table 7, the data obtained from release studies were fitted to different kinetic models (zero, first, Higuchi, Korsmeyer–Peppas and Hixson–Crowell models). Higher values of regression (*R*^2^) were observed for Korsmeyer–Peppas modeling (0.97 and 0.98 for the optimized formulation and F0, respectively) which were so close to the Higuchi diffusion model (0.96 and 0.97 for the optimized formulation and F0, respectively). Moreover, the *n* values were 0.56 and 0.52 for the optimized formulation and F0, respectively (the values lied between 0.45 and 0.89) indicating a non-Fickian release mechanism (anomalous diffusion). Hence, the QC release from both formulations was governed by diffusion, erosion, as well as degradation of the phospholipid matrix [73,74].

### 3.7. Ex Vivo Skin Permeation

Ex vivo skin permeation profiles of the optimized formula and F0 in comparison with the QC dispersion were carried out using Franz diffusion cells to assess the effect of the formulations on the diffusion of QC through full-thickness skin. Figure 8 displays the percentage of QC permeated through the skin versus the predetermined time intervals. After 8 h, the cumulative percentage of the permeated QC from the optimized formula was 58.6 ± 4.8% where it accounted for 47 ± 2.8% and 26.6 ± 1.9% of F0 and QC dispersion, respectively. It is clear that the percentages of permeation from phytosomal complexes were significantly (*p* < 0.05) higher than the QC dispersion. The optimized formulation (composed of an olive oil/PC ratio of 0.166, a QC/PC ratio of 1.95 and a surfactant concentration of 1.6%) showed the higher flux (12.52 ± 0.68 µg/cm^2^/h) and permeability coefficient (6.25 ± 0.50 × 10^−2^ cm^2^/h) when compared to other formulations. On the other hand, the flux of QC from phytosomal complexes was enhanced by 1.9 and 1.3 fold when compared to the QC suspension (Table 8). It should be noted that the anatomical skin structure is composed of three major layers: the epidermis, dermis and hypodermis [75] and associated appendages, such as hair follicles, sweat glands and nails [76]. In general, phospholipid-based systems such as phytosomes and liposomes can improve skin permeation through being fused/mixed with the lipid constitutions of the stratum corneum (the outermost layer in epidermis), allowing diffusion of drug molecules through the skin layers [35]. This fusion is leading to the creation of intercellular lipid lamellae and increases the motion of hydrophobic loaded molecules in the stratum corneum [77]. In addition, it can alter the fluidity of the skin barrier [76]. The lowest permeation characteristic of raw QC from the suspension might be attributed to its hydrophobic nature in raw form which precluded skin penetration. On the other hand, the optimized formulation showed higher permeation behavior when compared to F0 which might be attributed to several factors. First, olive oil (oleic acid rich vegetable oil) can improve the skin permeability by disrupting the cellular arrangement of the stratum corneum irreversibly, leading to better QC penetration through the different skin layers [78]. Moreover, the miscibility/mixing of olive oil with stratum corneum lipids can in turn endorse OC cutaneous penetration. Furthermore, the packing arrangement of fatty acids altered the fluidity of dermal lipids and aided QCs skin absorption [79]. Second, the addition of olive oil to phospholipids in the optimized formulation resulted in the decrease of consistency within the system (verified by DSC study) allowing a higher escape probability of QC. Third, PF-127 is a non-ionic surfactant and can interact with the skin leading to disorder in the dermal barrier in the stratum corneum and an enhanced QC cutaneous permeability [25]. These findings revealed that the optimized formulation offered a promising skin delivery system for QC.

Figure 9 displays the percentage of QC deposited in full-thickness skin at the end of the permeation study. After 8 h, the cumulative percentage of the deposited QC from the optimized formula was 31.4 ± 3.5% where it accounted for 21.3 ± 3.3% and 10.7 ± 1.8% of F0 and QC dispersion, respectively. It is clear that the percentage of accumulation from the optimized formula was significantly (*p* < 0.05) higher than both F0 and QC dispersion. This in turn could support the effectiveness of the optimized formulation for skin delivery of QC.

### 3.8. Skin Compliance 

In vivo skin compliance is an important element that should be estimated prior to recommending drug carrier as a dependable topical delivery system. However, the safety/toxicity features of the optimized formulation and F0, in comparison with the control group, were primarily appraised by visual inspection of any manifestations of skin irritation within 10 days of application. Both formulations did not exhibit any erythema or redness manifestation when compared to the untreated group. The erythema scores were zero for both groups indicating good skin safety and tolerance. As phospholipids (lecithin and hydrogenated lecithin) are generally reported as safe and well-tolerated by the skin [80,81], investigated formulations looked safe as they were phospholipid-based. Traul and coworkers found that a 30% dispersion of the phospholipids in medium-chain triglycerides is well-tolerated in dermal and cosmeceutical preparations [82]. On the other hand, a skin histopathology of different investigated skin samples was conducted to explore the influence of the optimized formulation and F0 on the skin structure. Figure 10A shows a photomicrograph of Hematoxylin and Eosin stained control of untreated skin. It shows a clear delineation between the epidermis and dermis, low neutrophil infiltration, normal arranged fine and coarse collagen fibers and a compact stratum corneum. The application of F0 (Figure 10B) shows an increase in the thickness of the stratum corneum. This increase was more prominent in optimized formulation-treated animals (Figure 10C) which might be ascribed to the occlusive effect offered by olive oil [83]. In addition, the stratum corneum of the skin specimens treated with the optimized formulation was found disrupted, to some extent, which was attributed to the action of both olive oil and phospholipid. It was reported that olive oil can reduce stratum corneum integrity [84] while phospholipid can change and fluidize the structure of the membrane barrier leading to the enhanced permeability of active agents [85]. 

## 4. Conclusions

The Box–Behnken design was used to optimize QC loaded olive oil containing phytosomal complexes prepared by solvent evaporation/anti-solvent precipitation techniques with satisfactory physicochemical properties for skin delivery. The optimized composition was found to be an olive oil/PC ratio of 0.166, a QC/PC ratio of 1.95 and a surfactant concentration of 1.6%. This composition showed reasonable particle diameter (206.7 nm), zeta potential value (−26.3 mV) and satisfying EE% (85.3%). The optimized formulation showed effective complexation between QC and PC in phytosomal complexes and a good stability over 3 months at room temperature. It also showed a biphasic pattern in in vitro release study. The skin permeation study showed that it could permeate more effectively with the optimized compared to the olive oil/surfactant free formulation and control (~1.9-fold and 1.3-fold, respectively). The optimized formulation was also well-tolerated by the skin and showed some histological changes which aided in skin permeation. Therefore, the optimized formulation was designated as a potential carrier for QC, as a natural bioactive agent, and to improve skin delivery, which could be further augmented by carrying out clinical experiments and scale-up studies in the future.

## Figures and Tables

**Figure 1 pharmaceutics-15-01124-f001:**
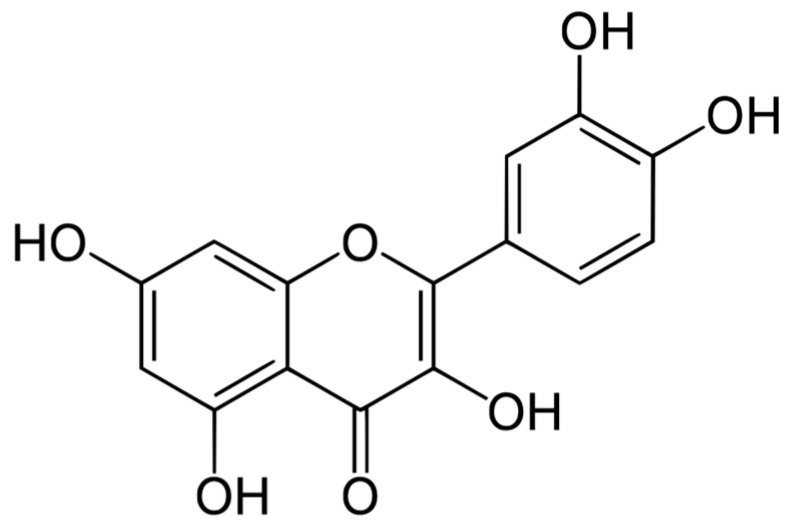
Chemical structure of quercetin.

**Figure 2 pharmaceutics-15-01124-f002:**
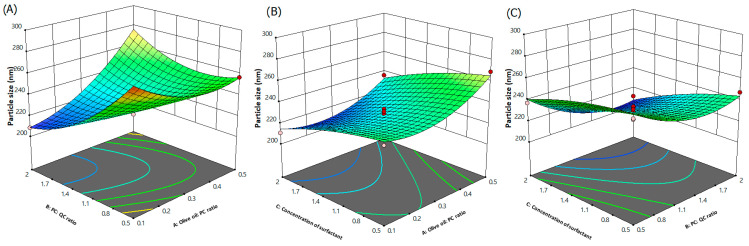
Three-dimensional response surface plots indicating the combined influence of: (**A**) olive oil/PC and QC/PC ratios, (**B**) olive oil/PC ratio and concentration of surfactant and (**C**) QC/PC ratio and concentration of surfactant on particle size.

**Figure 3 pharmaceutics-15-01124-f003:**
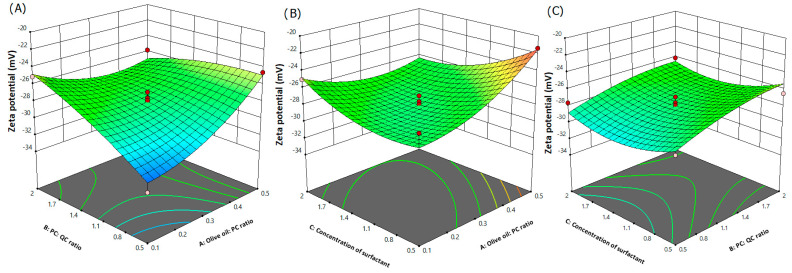
Three-dimensional response surface plots indicating the combined influence of: (**A**) olive oil/PC and QC/PC ratios, (**B**) olive oil/PC ratio and concentration of surfactant and (**C**) QC/PC ratio and concentration of surfactant on zeta potential.

**Figure 4 pharmaceutics-15-01124-f004:**
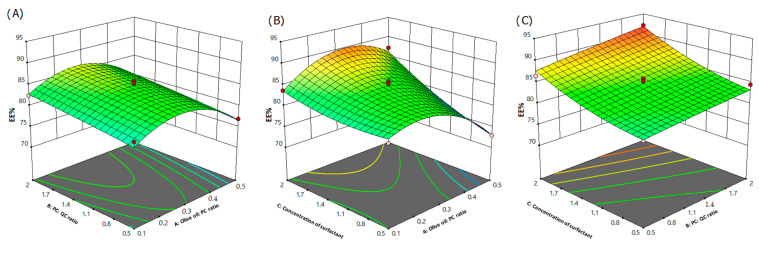
Three-dimensional response surface plots indicating the combined influence of: (**A**) olive oil/PC and QC/PC ratios, (**B**) olive oil/PC ratio and concentration of surfactant and (**C**) QC/PC ratio and concentration of surfactant on EE%.

**Figure 5 pharmaceutics-15-01124-f005:**
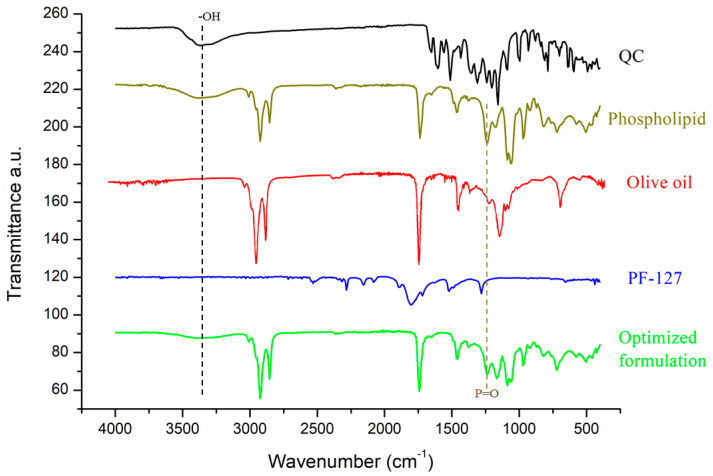
FT-IR spectra of QC, phospholipid, olive oil, PF-127 and optimized formulation.

**Figure 6 pharmaceutics-15-01124-f006:**
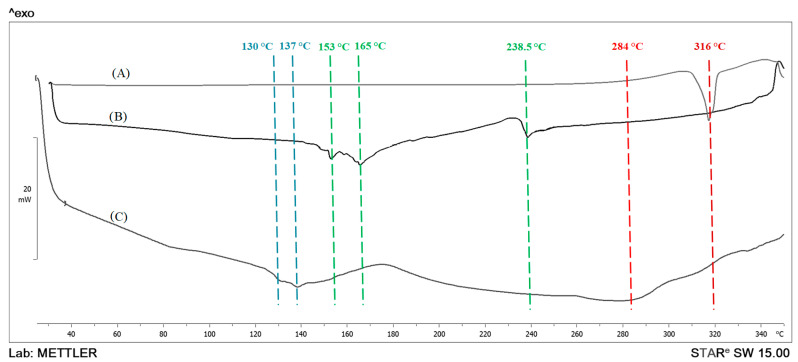
DSC thermograms of: (**A**) QC, (**B**) phospholipid and (**C**) optimized formulation.

**Figure 7 pharmaceutics-15-01124-f007:**
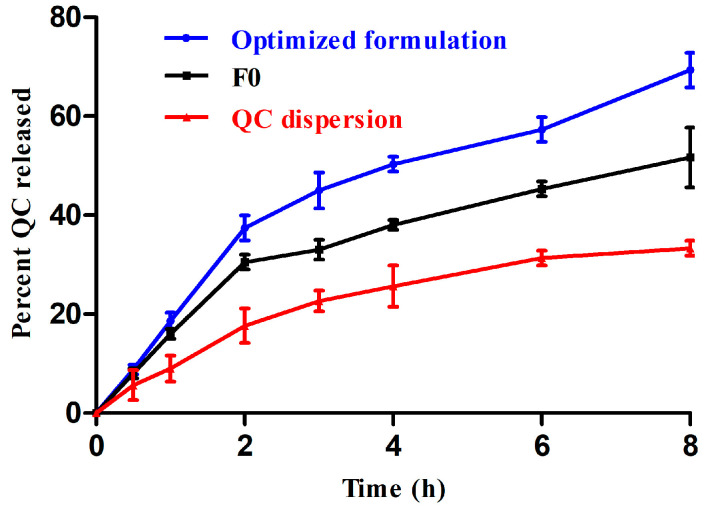
In vitro release patterns of QC from the optimized formula, F0 and QC dispersion (mean values ± SD, *n* = 3).

**Figure 8 pharmaceutics-15-01124-f008:**
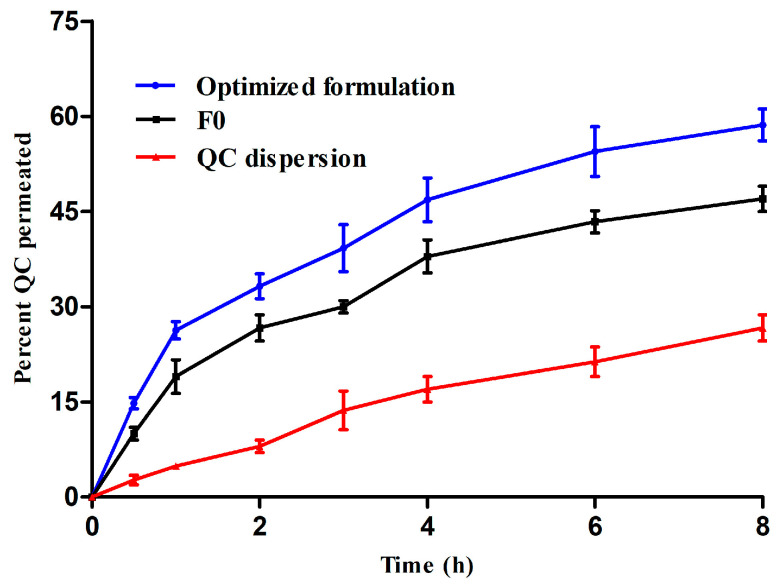
Skin permeation of QC from the optimized formula, F0 and QC dispersion (mean values ± SD, *n* = 3).

**Figure 9 pharmaceutics-15-01124-f009:**
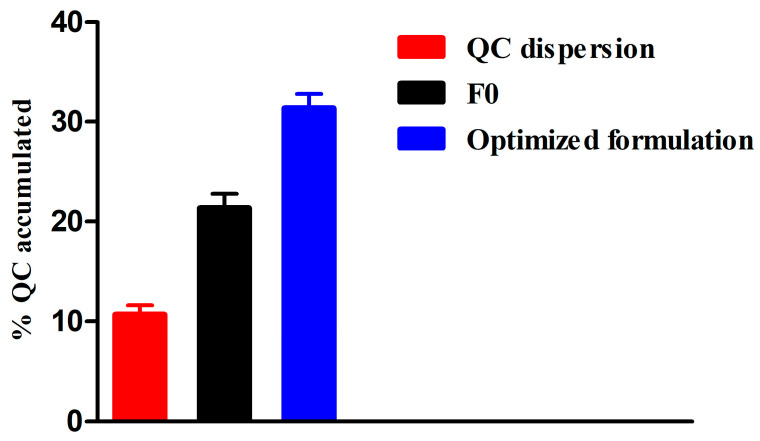
Cumulative amounts of QC deposited in the skin 8 h after application of the optimized formula, F0 and QC dispersion (mean values ± SD, *n* = 3).

**Figure 10 pharmaceutics-15-01124-f010:**
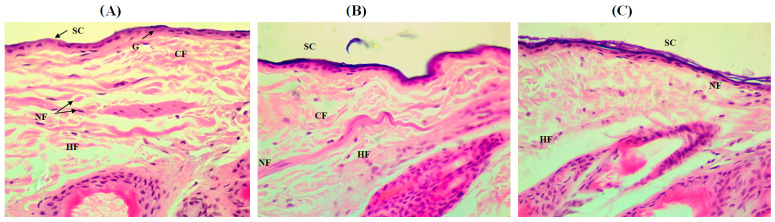
A Photomicrograph of (**A**) normal skin of control group, (**B**) skin specimen treated with F0, and (**C**) skin specimen treated with the optimized formulation. All samples were stained with standard Hematoxylin and Eosin. SC: stratum corneum, NF: neutrophil infiltration, G: granulosum, CF: Collagen Fibers, HF: Hair Follicle (Magnification 400×).

**Table 1 pharmaceutics-15-01124-t001:** Levels of independent variables and responses used in experimental design.

Factor	Process Parameter	Levels		
		Low	Medium	High
		(−1)	(0)	(1)
Independent variables				
A	Olive oil/PC ratio	1:10	1:5	1:2
B	QC/PC ratio	2:1	1:1	1:2
C	Concentration of surfactant	0.5%	1%	2%
Responses		Desired outcome		
Y_1_	Particle diameter	Decrease		
Y_2_	Surface charge	Increase		
Y_3_	Encapsulation efficiency %	Increase		

**Table 2 pharmaceutics-15-01124-t002:** Compositions of different formulations according to Box–Behnken design.

STD	Run	Olive Oil/PC Ratio	QC/PC Ratio	Concentration of Surfactant (% *w*/*v*)
8	1	0.5	1.25	2
13	2	0.3	1.25	1.25
9	3	0.3	0.5	0.5
14	4	0.3	1.25	1.25
11	5	0.3	0.5	2
10	6	0.3	2	0.5
5	7	0.1	1.25	0.5
7	8	0.1	1.25	2
16	9	0.3	1.25	1.25
3	10	0.1	2	1.25
2	11	0.5	0.5	1.25
17	12	0.3	1.25	1.25
12	13	0.3	2	2
1	14	0.1	0.5	1.25
4	15	0.5	2	1.25
15	16	0.3	1.25	1.25
6	17	0.5	1.25	0.5

**Table 3 pharmaceutics-15-01124-t003:** Summary of statistical models for the measured responses.

Particle Size								
Source	SD	*R* ^2^	Adjusted *R*^2^	Predicted *R*^2^	*p*-Value	Lack of Fit		Remark
						F-Value	*p*-Value	
Linear	17.66	0.4968	0.3806	0.0260	0.0263	20.23	0.0054	
2FI	13.74	0.7657	0.6252	0.1263	0.0462	13.77	0.0120	
Quadratic	6.45	0.9638	0.9173	0.5774	0.0032	3.13	0.1499	Suggested
Zeta Potential								
Linear	2.28	0.2844	0.1193	−0.4114	0.2115	9.37	0.0228	
2FI	1.80	0.6599	0.4559	−0.3876	0.0509	6.33	0.0478	
Quadratic	1.39	0.8579	0.6753	−0.8055	0.0900	4.51	0.0897	Suggested
EE%								
Linear	3.39	0.5122	0.3996	0.0379	0.0217	24.86	0.0037	
2FI	3.43	0.6171	0.3874	−0.7218	0.4688	29.13	0.0029	
Quadratic	1.20	0.9672	0.9251	0.5995	0.0004	3.77	0.1164	Suggested

**Table 4 pharmaceutics-15-01124-t004:** Particle size, zeta potential and EE% of different formulations.

Run	Particle Diameter (nm)	Zeta Potential (mV)	EE%
1	236	−27.8	87.2
2	221.6	−26.9	84.4
3	258.3	−28.6	80.7
4	228	−29.3	85.9
5	237.3	−27.6	86.5
6	247.3	−26.5	84.5
7	237.7	−26.3	80.6
8	210.8	−25	83.7
9	234	−28.1	85.4
10	208.5	−25.1	82.5
11	256.7	−24.6	76.9
12	231.5	−27.9	83.9
13	210.9	−26.1	92.1
14	287.9	−33	80.8
15	268.9	−25.8	78.9
16	229.4	−27.6	84.5
17	268.9	−21.4	72.9

**Table 5 pharmaceutics-15-01124-t005:** Predicted and actual values of independent variables and responses for the optimized formulation.

Formula	Independent Variables			Dependent Variables			Desirability
	Level of Factor A	Level of Factor B	Level of Factor C	Particle Size (nm)	Zeta Potential (mV)	EE%	
Predicted	0.1664	1.95	1.61	203.635	−25.3123	86.427	1
Actual	0.1664	1.95	1.61	206.7	−26.3	85.3	---

**Table 6 pharmaceutics-15-01124-t006:** Stability studies of optimized formula and F0 (mean ± DS; *n* = 3).

Assessment Criteria	Storage Condition	Freshly Prepared		1st Month		2nd Month		3rd Month	
		F0	Optimized Formulation	F0	Optimized Formulation	F0	Optimized Formulation	F0	Optimized Formulation
Particle size (nm)	At 25 ± 2 °C	211.8 ± 21.6	206.7 ± 15.8	229.9 ± 11.5	210.3 ± 11.5	238.1 ± 17.6	217.8 ± 22.1	256.8 ± 15.6	228.1 ± 12.4
	At 4 °C	211.8 ± 21.6	211.8 ± 18.8	237.6 ± 18.6	216.8 ± 14.1	259.4 ± 16.2	231.6 ± 11.7	295.7 ± 18.7	243.8 ± 16.4
PI	At 25 ± 2 °C	0.23 ± 0.02	0.25 ± 0.03	0.24 ± 0.01	0.25 ± 0.02	0.26 ± 0.05	0.26 ± 0.04	0.28 ± 0.06	0.26 ± 0.02
	At 4 °C	0.23 ± 0.02	0.25 ± 0.03	0.25 ± 0.03	0.27 ± 0.05	0.31 ± 0.06	0.28 ± 0.05	0.36 ± 0.05	0.30 ± 0.04
Zeta potential (mV)	At 25 ± 2 °C	−35.5 ± 4.8	−26.3 ± 7.1	−33.6 ± 9.5	−27.1 ± 8.9	−32.6 ± 1.9	−25.2 ± 5.1	−32.9 ± 6.9	−26.2 ± 3.6
	At 4 °C	−35.5 ± 4.8	−26.3 ± 7.1	−32.1 ± 7.3	−25.3 ± 8.1	−33.5 ± 7.3	−24.9 ± 6.2	−32.3 ± 3.8	−25.9 ± 7.5
EE%	At 25 ± 2 °C	85.1 ± 7.9	85.3 ± 12.5	84.6 ± 6.5	86.7 ± 7.5	83.4 ± 11.5	85.7 ± 4.5	81.1 ± 10.5	84.7 ± 13.3
	At 4 °C	85.1 ± 7.9	85.3 ± 12.5	83.2 ± 8.5	84.3 ± 10.9	81.9 ± 5.8	83.6 ± 7.1	80.4 ± 13.7	83.3 ± 11.8

**Table 7 pharmaceutics-15-01124-t007:** Fitting of release data with different kinetic models.

Code	Zero Order Kinetics			First Order Kinetics			Higuchi Model			Hixson-Crowell			Korsmeyer-Peppas		
	*R* ^2^	*t* _1/2_	*K*	*R* ^2^	*t* _1/2_	*K*	*R* ^2^	*t* _1/2_	*K*	*R* ^2^	*t* _1/2_	*K*	*R* ^2^	*t* _1/2_	*K*
Optimized formulation	0.80	4.88	10.23	0.95	4.05	0.17	0.96	4.26	24.2	0.92	4.2	0.049	0.97	4.3	22.08
F0	0.75	6.4	7.8	0.89	6.2	0.11	0.97	7.2	18.6	0.86	6.2	0.03	0.98	7.1	17.9

**Table 8 pharmaceutics-15-01124-t008:** Skin permeation parameters of different investigated formulations.

Formula Code	Flux, Jss (µg/cm^2^/h)	Permeability Coefficient Kp × 10^−2^ (cm^2^/h)	Enhancement Ratio (ER)
Optimized formulation	12.52 ± 0.68	6.25 ± 0.50	1.9
F0	8.83 ± 0.63	4.41 ± 0.25	1.3
QC dispersion	6.71 ± 0.42	3.35 ± 0.37	---

## Data Availability

Not applicable.

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
