# Peer review of "Development of Olive Oil Containing Phytosomal Nanocomplex for Improving Skin Delivery of Quercetin: Formulation Design Optimization, In Vitro and Ex Vivo Appraisals"

_pharmaceutics, 2023, doi:10.3390/pharmaceutics15041124_

Round 1

Reviewer 1 Report (New Reviewer)

The article presented by Hendawy et al. is focused on the synthesis of olive oil/phytosomal nanocarriers for improving quercetin skin delivery. The results are interesting, and although they do not use very novel synthesis methods, the results are interesting, obtaining nanocarriers with a good % of encapsulation and stability from the physicochemical point of view. In addition, the nanocarrier obtained could allow the use of quercetin, also very interesting from the pharmacological point of view. This manuscript is well written, and experiments were carefully conducted, only one comment:
* Stability tests over time are carried out only by studies about of physical chemical parameters. Sometimes, events occur in the nanoparticles which are not reflected in these parameters, such as drug degradation. It would also be interesting to perform biological assays of in vitro release and ex vivo skin permeation with quercetin in Olive oil/phytosomal complexes loaded with QC after 1, 2 and 3 months of initial synthesis.

The rest of the article is well-written and there are no errors or missing experiments.

Author Response

Comments and Suggestions for Authors

The article presented by Hendawy et al. is focused on the synthesis of olive oil/phytosomal nanocarriers for improving quercetin skin delivery. The results are interesting, and although they do not use very novel synthesis methods, the results are interesting, obtaining nanocarriers with a good % of encapsulation and stability from the physicochemical point of view. In addition, the nanocarrier obtained could allow the use of quercetin, also very interesting from the pharmacological point of view. This manuscript is well written, and experiments were carefully conducted, only one comment:
* Stability tests over time are carried out only by studies about of physical chemical parameters. Sometimes, events occur in the nanoparticles which are not reflected in these parameters, such as drug degradation. It would also be interesting to perform biological assays of in vitro release and ex vivo skin permeation with quercetin in Olive oil/phytosomal complexes loaded with QC after 1, 2 and 3 months of initial synthesis.

Authors’ reply:

Authors appreciate the learned referee for this valuable comment. Actually, colloidal systems are, in general, suffering from stability problems particularly that related to physicochemical properties such as particle size growth, multi-modal particle distribution and decrease in zeta potential value. As the developed system (olive oil containing phytosomal complex) has not been discussed in literature, we performed stability studies in two temperatures; room and refrigerating in order to investigate the most suitable storage conditions. On the other hand, we did not neglect the percentage of quercetin encapsulated in the system and measured EE% during 3 months. In vitro release and ex vivo skin permeability were performed for 8h which was a short period to produce hydrolytic degradation of quercetin. On the other hand, determination of quercetin amount was carried out using high sensitive technique (UPLC).  

The rest of the article is well-written and there are no errors or missing experiments.

Authors’ reply:

Authors are thankful to the learned referee.

Reviewer 2 Report (New Reviewer)

Hendawy and co-authors presented an original paper entitled: “Development of olive oil containing phytosomal nanocomplex for improving skin delivery of quercetin: formulation design optimization, in vitro and ex vivo appraisals” for the publication on Pharmaceutics. After a deep revision of the manuscript, this Reviewer appreciates the idea on which the work is based but does not consider the paper suitable for publication in this current form. Major revisions are needed. In general, poor attention has been paid to the writing of the manuscript and additional experiments are required.

Specific comments:

-          the title is convoluted and too long

-          more references are needed, above all in introduction section

-          many typos are present in the entire manuscript (for example, in 2.1 section, in 2.5 section, etc)

-          2.2 section: This Reviewer supposes that the author for the preparation have used not a simply stirring but a rotational system similar to rotavapor for the thin layer formation. If not, explain how a thin film can be obtained using the method described. Please better explain

-          2.2 section: “Then, the thin film was dissolved in 10 ml of n-hexane with stirring in the fume hood until complete removal of n-hexane which was verified by weighing the mixture prior solvent addition and after its evaporation (22).” what is the meaning of this passage?

-          2.4 section: References are needed to justify the DLS technique. The authors can use this https://doi.org/10.3390/pharmaceutics12111127

-          2.8 section: the methods were written superficially, hopefully they were at least conducted in a better way. Some important information is missing, such as the temperature at which the experiments were conducted. Was an ambient temperature maintained? skin temperature? or body temperature? Moreover, the authors studied the in vitro release of QC dispersion, what is meant by dispersion? aqueous? Hydroalcoholic solution?

-          2.9 section: please revised the first paragraph.

-          2.9 section: please better explain the composition of Ringer solution

-          2.9 section: why in this case the composition of the receptor has changed, compared to in vitro release studies? it would have been appropriate (for the reproducibility and reliability of the data, above all considering the lipophilicity of the drug) to use the same medium.

-          2.10 section: how many animals were used? what are the ethical protocol codes for the use of live animals?

-          2.11 section: the authors described the methods used for statistical analysis, but unfortunately in no passage of the text and in no figure is the statistical analysis present. please take care of everything.

-          3.1 section: a reference is needed to justify the first sentence.

-          3.1. section: What characteristics of phytosomes suggest a good ability to deliver drugs through the skin?

-          3.5 section: “optimized formulation”, it would be appropriate to specify what the formulation is and its composition. It is inconvenient for the reader to keep the sample in mind or go back through the text.

-          Table 6: Are there any statistically significant differences?

-          3.7 section: Have you also evaluated the quantity sequestered from the skin tissue? if it has not been done, it would be advisable to do so for completeness of the data.

-          3.8 section: “to the occlusive effect offered by olive oil which could increase skin hydration by preventing transepider-mal water loss” to demonstrate this, it is not enough to put a reference, but it would be advisable to carry out the same test by testing the olive oil alone on the skin.

-          3.8 section: this could lead to damage. have you evaluated whether the destruction is reversible or irreversible? even in this case, if they are not proven, they remain only guesses.

Author Response

Comments and Suggestions for Authors

Hendawy and co-authors presented an original paper entitled: “Development of olive oil containing phytosomal nanocomplex for improving skin delivery of quercetin: formulation design optimization, in vitro and ex vivo appraisals” for the publication on Pharmaceutics. After a deep revision of the manuscript, this Reviewer appreciates the idea on which the work is based but does not consider the paper suitable for publication in this current form. Major revisions are needed. In general, poor attention has been paid to the writing of the manuscript and additional experiments are required.

Specific comments:

-          the title is convoluted and too long

Authors’ reply:

  1. Title has been shortened.

-          more references are needed, above all in introduction section

Authors’ reply:

  1. More references have been added to Introduction-section. All changes have been yellow highlighted.

-          many typos are present in the entire manuscript (for example, in 2.1 section, in 2.5 section, etc)

Authors’ reply:

  1. We have gone through the manuscript and revised all typos. All changes have been yellow highlighted.

-          2.2 section: This Reviewer supposes that the author for the preparation have used not a simply stirring but a rotational system similar to rotavapor for the thin layer formation. If not, explain how a thin film can be obtained using the method described. Please better explain

Authors’ reply:

We used magnetic stirring for evaporation of organic solvent. The statement has been paraphrased to be more comprehensive. All changes have been yellow highlighted.  

-          2.2 section: “Then, the thin film was dissolved in 10 ml of n-hexane with stirring in the fume hood until complete removal of n-hexane which was verified by weighing the mixture prior solvent addition and after its evaporation (22).” what is the meaning of this passage?

Authors’ reply:

  1. We have divided the sentence into two statements. It is now “Then, the thin film was dissolved in 10 ml of n-hexane using continuous stirring in the fume hood. Complete removal of n-hexane was verified by weighing the mixture prior solvent addition and after its evaporation (22).” All changes have been yellow highlighted.

-          2.4 section: References are needed to justify the DLS technique. The authors can use this https://doi.org/10.3390/pharmaceutics12111127

Authors’ reply:

  1. Recommended reference has been added.

-          2.8 section: the methods were written superficially, hopefully they were at least conducted in a better way. Some important information is missing, such as the temperature at which the experiments were conducted. Was an ambient temperature maintained? skin temperature? or body temperature? Moreover, the authors studied the in vitro release of QC dispersion, what is meant by dispersion? aqueous? Hydroalcoholic solution?

Authors’ reply:

  1. Temperature was kept during the experiment at 37 °C ±1. QC dispersion was prepared by dispersion in 0.5% methylcellulose. We have added such details in the section 2.8. Some statements have been paraphrased to be more clear. All changes have been yellow highlighted.

-          2.9 section: please revised the first paragraph.

Authors’ reply:

  1. The statement has been divided to be more clear. It becomes “Ex vivo skin permeation has been carried out for optimized formulation and F0 in comparison with QC dispersion as a control. All formulations were used in a dose equivalent to (20 mg).” All changes have been yellow highlighted.

-          2.9 section: please better explain the composition of Ringer solution

Authors’ reply:

  1. The composition has been added.

-          2.9 section: why in this case the composition of the receptor has changed, compared to in vitro release studies? it would have been appropriate (for the reproducibility and reliability of the data, above all considering the lipophilicity of the drug) to use the same medium.

Authors’ reply:

  1. The receptor medium in section- “2.9. Ex vivo skin permeation” was incorrectly entered. We used Hydroalcoholic solution (composed of phoshate buffered saline/ethyl alcohol (70:30 v/v; pH = 7.4)) in both experiments. We have corrected the mistake. However, the solubility of QC was predetermined in receptor compartment as 3.9±0.07 mg/ml.

.

-         2.10 section: how many animals were used? what are the ethical protocol codes for the use of live animals?

Authors’ reply:

Number of animals has been added. Ethical code for the experiments in the study was mentioned in section 2.9.  

-          2.11 section: the authors described the methods used for statistical analysis, but unfortunately in no passage of the text and in no figure is the statistical analysis present. please take care of everything.

Authors’ reply:

  1. Statistical data has been added through the manuscript. All changes have been yellow highlighted.

-          3.1 section: a reference is needed to justify the first sentence.

Authors’ reply:

This is a starting statement. The system has not previously used for skin delivery. So, we cannot add reference to the statement. However, we paraphrased the statement not to lead to misunderstanding.

-          3.1. section: What characteristics of phytosomes suggest a good ability to deliver drugs through the skin?

Authors’ reply:

As the developed system (olive oil containing phytosomal complex) has not been discussed in literature for skin delivery, the mechanism by which it can improve skin delivery was aligned to the components of the system (phospholipid and olive oil). Details and supporting studies were mentioned in section 3.7.

-          3.5 section: “optimized formulation”, it would be appropriate to specify what the formulation is and its composition. It is inconvenient for the reader to keep the sample in mind or go back through the text.

Authors’ reply:

  1. The composition of optimized formulation has been added in many sites in the manuscript. . All changes have been yellow highlighted.

-          Table 6: Are there any statistically significant differences?

Authors’ reply:

Statistical significances (based on P value) have been added in the text. All changes have been yellow highlighted.  

-          3.7 section: Have you also evaluated the quantity sequestered from the skin tissue? if it has not been done, it would be advisable to do so for completeness of the data.

Authors’ reply:

  1. New data has been added. We have determined QC skin deposition after the end of permeation experiment (8h). All changes have been yellow highlighted.

-          3.8 section: “to the occlusive effect offered by olive oil which could increase skin hydration by preventing transepider-mal water loss” to demonstrate this, it is not enough to put a reference, but it would be advisable to carry out the same test by testing the olive oil alone on the skin.

Authors’ reply:

  1. Details have been removed from the statement.

-          3.8 section: this could lead to damage. have you evaluated whether the destruction is reversible or irreversible? even in this case, if they are not proven, they remain only guesses.

Authors’ reply:

Actually, we have never mentioned in the section that application of F0 or optimized formulation can lead to destruction of stratum corneum. We used “disrupted” expression to illustrate the condition of the stratum corneum of the skin after application which is very familiar term in that regard. On the other hand, our discussion was supported by relevant references (84 and 85).

Round 2

Reviewer 2 Report (New Reviewer)

The authors have responded more or less satisfactorily to my requests. I suggest revising the English though.

This manuscript is a resubmission of an earlier submission. The following is a list of the peer review reports and author responses from that submission.

Round 1

Reviewer 1 Report

This manuscript described the development of hyaluronate-based olive oil/phytosomal nanoparticles prepared by a solvent evaporation/anti-solvent precipitation technique for the delivery of a flavonoid-structure drug, quercetin. Different preclinical factors were systematically studied. The paper could be potentially helpful to the scientific community but is not currently at that state of completion. The manuscript is well organized, but the novelty of this work is not so clear. The importance of the research fails to be directly and clearly expressed, and some results in this paper were presented without sufficient discussion. Some procedures of the experiment were poorly described, and the lack of accurate reference between the result discussion and its corresponding Figures/Tables caused confusion when reading the paper. Thus, major revisions are required to strengthen this manuscript before it can be accepted by Pharmaceutics.

The paper focuses a great deal of attention on analysis that have very little biological relevance. As a result, it is hard to know what new information is brought to the field by this study.  With little new information relayed how this might help the target biological effect, one is left to think this study was purely an analytical study without much consideration for its final application. Please note, the reviewer is not saying that this article might not be helpful to the community, just that at this point it is not yet helpful nor thorough enough to be helpful and to push science relevant to the readership of Pharmaceutics.

General considerations:

·       What is new in this work as compared to other works? In the Introduction part, please focus on deliver the novelty of this work.  So please rewrite the Introduction part.

·       The figures (e.g., figures in Figures 1, 2 and 3) should be labelled with (a), (b), (c), etc to make each of the figures better understood, with a brief explanation in the footnote about each of the graphs.

·       There are quite many typos and grammar mistakes. Some examples are listed below. In addition, the editing of the manuscript could have been improved. The authors should carefully revise and re-edit the manuscript before submitting the new version.

·       One large issue is that the methods section has poorly information that will make these studies reproducible, and without that critical information, the reviewer also cannot deeply interpret if these studies were appropriately conducted.  Therefore, the manuscript should be revised and more detailed information on the methodology section should be provided.

·       The safety of these systems at the ocular level has not been studied. This feature should be considered together with the study of the efficacy of the system, since an effective but toxic system would not offer, a priori, therapeutic usefulness. It is advisable to perform safety studies using organotypic models (such as BCOP or HET-CAM test) to make an initial assessment of these systems in terms of safety for ocular administration.

·       Olive oil is associated with oxidation and rancidity processes, largely conditioning the stability of the drug delivery systems of which it is a part. What is the reason for having selected this component and not another type of lipid with similar properties but in which these processes are minimized? Have other lipids been previously tested by the authors?

·       Neither particle production yield nor particle loading capacity has been studied, so quantitative analysis of these systems falls slightly poor.

·       No studies have been carried out in terms of the nanoparticle's morphology, so it is being assumed that these nanoparticles showed a spherical structure with a regular surface. It would be advisable to perform the morphology analyses by SEM and TEM, since the morphology significantly conditions the bioavailability and ocular penetration of these systems.

·       Since these systems have drug entrapment due to charge interaction, a stability study at pH and ionic strength should be performed. Has the pH of the formulation been tested once it has been prepared? How does the use of buffer solutions affect the sample stability?

Specific comments:

·       Section 2.2. Evaporation of n-hexane was determined by weight difference, but traces that could significantly affect the ocular tissue due to the solvent cytotoxicity could remain in the final formulation. A TGA (or similar) should be performed to check that the solvent had been completely removed (or that the remaining concentration was below the established limits).

·       Section 2.2. No sonication conditions were included. Sonication generates heat, which may condition the stability formulation or any of its components. The authors should give some insights about this observation.

·       Section 2.4. DLS subsets for size determination, PDI and Z-potential are not specified. They should at least be included: temperature stabilization time, analysis runs for each formulation, determinations per run for each sample.

·       Section 2.4. What is the composition of the 0.22 micron filter used to filter the supernatant? Filter composition may condition the amount of drug entrapped during the filtration process.

·       Section 2.4. Has the HPLC method been previously validated? If so, it would be advisable to include information about linear behaviour (correlation coefficient), concentration interval, accuracy, LOQ and LOD, among others.

·       Section 2.6. What was the sample preparation process for this study? Were the nanoparticles kept in suspension during the study or kept lyophilized and reconstituted at the time of measurement? If lyophilized, did the authors employ any type of cryoprotectant for the freeze-drying of nanoparticles? The authors should give some insights about this observation.

·       Section 2.6. It is stated that the F0 formulation is prepared either without olive oil or without Pluronic F127, so it is understood that two different types of F0 formulation (F0a and F0b) are used. If so, in which assays is one used and in which the other?

·       Section 2.7. Why is PBS/Ethyl Alcohol 70:30 solution used, instead of using PBS exclusively?

·       Section 2.7. What type of diffusion cells were used? Horizontal or vertical Franz diffusion cells?

·       Section 2.8. Why have the size, PDI and Z-potential of HA-coated NPs not been evaluated if they are really intended to be the final formulation? This information should be included in the new version of the manuscript.

·       Section 2.9. Why was this concentration of quercetin taken as the study concentration, and is there any evidence on which this choice was based?

·       Section 2.9. Tween 80 was used in the ex vivo transcorneal permeation study but was not included in the materials section. What is the rationale for using that concentration of Tween in the donor portion along with the buffer solution?

·       Section 2.9. Why are different volumes used in the donor and receptor chambers of this assay with respect to the in vitro assay. Are different Franz cell types used?

·       Section 2.10. No post hoc analysis was applied in the different studies. Multiple comparisons test (Tukey or Bonferroni test, among others) should be applied among formulations to deeply analysis the preclinical data.

·       Section 3.2. PDI is not considered. PDI data should be included along with the other data, since it provides information on the homogeneity of the nanoparticle populations and, subsequently, about the reproducibility of the processing method.

·       Section 3.1.4. The units of measurement of the ratios and surfactant concentration were not indicated.

·       Section 3.3. Has the reason for this phenomenon of size increase of the F0 formulation been studied? The Z potential values seem to be conserved independently of time and temperature, so the repulsion phenomenon between particles should be preserved. The authors should give some insights about this observation.

·       Section 3.4. Why was the in vitro release study carried out for only 8h? Since these are controlled-release dosage forms, it would be ideal to evaluate the release profile over longer time periods to assess the possibility of reducing the frequency of administration, while preserving efficacy, and thus increase patient adherence to treatment. The controlled release performance of nanoparticles is related to their dispersion state, particle size and morphology.  How to ensure stable controlled release performance when these three parameters change?

·       Section 3.4. The F0 formulation generally shows a longer half-life, in some cases almost double that of the optimized formulation. Should this be considered a disadvantage?

·       Section 3.4. A quercetin suspension was not included in the release study as a second control formulation. The authors should give some insights about this observation.

Tables and figures:

·       Table 2. It would be appropriate to separate the data related to the nanoparticles composition and the results obtained after the systems elaboration in two different tables. Each table should be placed in the corresponding section (methodology and results, respectively).

·       Table 2. The units of measurement of the ratios and surfactant concentration were not shown.

·       Table 5. The data in this table should be graphically represented to improve the reader's understanding of the data.

Minor changes:

·       There are formatting errors in the organization of the results section (sections 3.1.1 and successive sections after section 3.2).

Author Response

This manuscript described the development of hyaluronate-based olive oil/phytosomal nanoparticles prepared by a solvent evaporation/anti-solvent precipitation technique for the delivery of a flavonoid-structure drug, quercetin. Different preclinical factors were systematically studied. The paper could be potentially helpful to the scientific community but is not currently at that state of completion. The manuscript is well organized, but the novelty of this work is not so clear. The importance of the research fails to be directly and clearly expressed, and some results in this paper were presented without sufficient discussion. Some procedures of the experiment were poorly described, and the lack of accurate reference between the result discussion and its corresponding Figures/Tables caused confusion when reading the paper. Thus, major revisions are required to strengthen this manuscript before it can be accepted by Pharmaceutics.

Authors’ reply:

Although the work seems a routine in physicochemical evaluations and ex vivo study, the design of the formulation, in our opinion, can lead to new discovery. However, and to best of our knowledge, none of the previous studies in the literature have performed a systematic study of the simultaneous influence of olive oil/phospholipid ratio, quercetin/phospholipid ratio, and concentration of surfactant on physicochemical behaviour of the developed phytosomes in order to improve ocular delivery. So, we have added this statement in “Introduction “-section. All changes have been yellow highlighted in the manuscript.

Actually, we never mention in the text that using phytosomes for ocular delivery of quercetin would be a novel idea, but it may lead to new discovery based on the proper investigation and interpretation. Our aim has from the beginning been to exploit the benefits olive oil/pluronic containing phytosomes (infrequently used in ocular formulations) for ocular delivery of quercetin to provide a promising approach to enhance its antioxidant and anti-inflammatory effects. Studying the factors influencing the formulation attributes is a major challenge to attain the aim.

The paper focuses a great deal of attention on analysis that have very little biological relevance. As a result, it is hard to know what new information is brought to the field by this study.  With little new information relayed how this might help the target biological effect, one is left to think this study was purely an analytical study without much consideration for its final application. Please note, the reviewer is not saying that this article might not be helpful to the community, just that at this point it is not yet helpful nor thorough enough to be helpful and to push science relevant to the readership of Pharmaceutics.

Authors’ reply:

Regarding biological evaluation, the authors are considering the current work as preliminary study. We primarily did the ex vivo corneal permeation studies for this manuscript to investigate the capability of the optimized formulation to diffuse through the cornea.

Naturally, in vivo results could improve our understanding of the behavior of the optimized formulation and prove its pharmacodynamic effects and safety, but as animal experiments are ethically questionable we have been very cautious to apply these in our research so far. However, we have performed an in vivo pilot study with only one rat each for the optimized formulation to check the safety. This result shows very minimal ocular toxicity of the optimized formulation (please see the figure below). So, this work will be completed in “Part-II” which will include cytotoxicity, in vivo ocular safety/toxicity, in vivo anti-inflammatory and antioxidant efficacy, deposition of quercetin in different corneal layer and corneal kinetics. We referred to this issue in “Conclusion”- section in the last statement as follows; “Therefore, HAOPC designated as a potential carrier for QC, as a natural bioactive agent, to improve its ocular delivery which could be further augmented by carrying out clinical experiments and scale-up studies in the future”.

General considerations:

  • What is new in this work as compared to other works? In the Introduction part, please focus on deliver the novelty of this work.  So please rewrite the Introduction part.

Authors’ reply:

It is correct that the manuscript looks like a routine in physicochemical evaluations and ex vivo study, the design of the formulation, in our opinion, can lead to new discovery. However, and to best of our knowledge, none of the previous studies in the literature have performed a systematic study of the simultaneous influence of olive oil/phospholipid ratio, quercetin/phospholipid ratio, and concentration of surfactant on physicochemical behaviour of the developed phytosomes in order to improve ocular delivery. So and as mentioned above, we have added this statement in “Introduction “-section. On the other hand, we have never mentioned in the text that using phytosomes for ocular delivery of quercetin would be a novel idea, but it may lead to new discovery based on the proper investigation and interpretation. Our aim has from the beginning been to exploit the benefits olive oil/pluronic containing phytosomes (infrequently used in ocular formulations) for ocular delivery of quercetin to provide a promising approach to enhance its antioxidant and anti-inflammatory effects. Studying the factors influencing the formulation attributes is a major challenge to attain the aim.

  • The figures (e.g., figures in Figures 1, 2 and 3) should be labelled with (a), (b), (c), etc to make each of the figures better understood, with a brief explanation in the footnote about each of the graphs.

Authors’ reply:

  1. The changes have been done. All changes have been yellow highlighted in the manuscript.

  • There are quite many typos and grammar mistakes. Some examples are listed below. In addition, the editing of the manuscript could have been improved. The authors should carefully revise and re-edit the manuscript before submitting the new version.

Authors’ reply:

We have carefully gone through the manuscript and it has been edited for typos and grammar mistakes. All changes have been yellow highlighted in the manuscript.

  • One large issue is that the methods section has poorly information that will make these studies reproducible, and without that critical information, the reviewer also cannot deeply interpret if these studies were appropriately conducted.  Therefore, the manuscript should be revised and more detailed information on the methodology section should be provided.

Authors’ reply:

More details were added methodologies with citing relevant references to be more clear.

  • The safety of these systems at the ocular level has not been studied. This feature should be considered together with the study of the efficacy of the system, since an effective but toxic system would not offer, a priori, therapeutic usefulness. It is advisable to perform safety studies using organotypic models (such as BCOP or HET-CAM test) to make an initial assessment of these systems in terms of safety for ocular administration.

Authors’ reply:

In vivo experiments, including cytotoxicity, in vivo ocular safety/toxicity, in vivo anti-inflammatory and antioxidant efficacy, deposition of quercetin in different corneal layer and corneal kinetics, are planned to be carried out in the future in the second part of our work. As we mentioned above, animal experiments are ethically questionable. So, we will perform full experiments after obtaining ethical approval. However, we have performed an in vivo pilot study with only one rat each for the optimized formulation to check the safety. This result shows very minimal ocular toxicity of the optimized formulation (please see the figure above). So, this work will be completed in “Part-II” and the recommended studies would be performed. On the other hand, above mentioned safety studies using organotypic models (such as BCOP or HET-CAM test) are not available in our lab.

  • Olive oil is associated with oxidation and rancidity processes, largely conditioning the stability of the drug delivery systems of which it is a part. What is the reason for having selected this component and not another type of lipid with similar properties but in which these processes are minimized? Have other lipids been previously tested by the authors?

Authors’ reply:

Phospholipid complexation can improve the oxidative stability of such oils (please see Maria del PilarGarcia-Mendoza et al., Food Chemistry, 2021,341, 128234). In addition, olive oil is a vegetable oil which was used in preparation of ophthalmic nanoemulsion (please see Agnieszka Gawin-MikoÅ‚ajewicz et al., Mol. Pharmaceutics 2021, 18, 3719−3740).  Furthermore, olive oil is rich in oleic, linoleic and α-linolenic acids. Oleic acid improves the ocular drug delivery of both hydrophilic and lipophilic compounds (please see Xiang-Chun Gao et al., 2014, Current Eye Research, Early Online, 1–8). Linoleic and α-linolenic play a role in stabilizing biologic membranes by creating physical properties that are optimal for the transport of substances across the membrane and for the biochemical reactions occurring in the membrane. Olive oil is also cultivated and harvested in Aljouf region, Saudi Arabia, so we can get high grades (extra virgin) of such oil easily.

  • Neither particle production yield nor particle loading capacity has been studied, so quantitative analysis of these systems falls slightly poor.

Authors’ reply:

As the time is limited (only 10 days for the response to reviewer), we will measure particle production yield and particle loading capacity for the optimized formulation in part-II work.

  • No studies have been carried out in terms of the nanoparticle's morphology, so it is being assumed that these nanoparticles showed a spherical structure with a regular surface. It would be advisable to perform the morphology analyses by SEM and TEM, since the morphology significantly conditions the bioavailability and ocular penetration of these systems.

Authors’ reply:

We are planning to do such evaluations in part-II work in parallel to in vivo evaluation.

  • Since these systems have drug entrapment due to charge interaction, a stability study at pH and ionic strength should be performed. Has the pH of the formulation been tested once it has been prepared? How does the use of buffer solutions affect the sample stability?

 Authors’ reply:

In this work, we used acetone as reaction solvent because it was reported to organize and not to disrupt the proton exchange process. We also used n-hexane antisolvent to separate phytosomal complexes by precipitation out from the organic solvent. When the QC in a defined quantity is dissolved in acetone with phospholipid, interaction between the polar heads of phospholipid would interact with QC to form phytosomal complex by weak interactions. By continuing, more complexes would be formed. So, the complex between QC and phospholipid took place before addition of hydrating solution.

Specific comments:

  • Section 2.2. Evaporation of n-hexane was determined by weight difference, but traces that could significantly affect the ocular tissue due to the solvent cytotoxicity could remain in the final formulation. A TGA (or similar) should be performed to check that the solvent had been completely removed (or that the remaining concentration was below the established limits).

Authors’ reply:

Of course organic solvent should be completely eliminated for formulation or at least existed in very low level according to established limits. In that regard, we depended on previously published work (Komeil et al., International Journal of Pharmaceutics 601 (2021) 120564) in which net weight of the prepared phytosome was determined before and after evaporation of organic solvent to ensure the complete evaporation. We have added this reference in Methodology part at the relevant site.

  • Section 2.2. No sonication conditions were included. Sonication generates heat, which may condition the stability formulation or any of its components. The authors should give some insights about this observation.

Authors’ reply:

We used probe sonication during fabrication of phytosomes (Please see section 2.2. Fabrication of olive oil/phytosomal complexes). In addition, relevant reference for the preparation method was included in the same section.

  • Section 2.4. DLS subsets for size determination, PDI and Z-potential are not specified. They should at least be included: temperature stabilization time, analysis runs for each formulation, determinations per run for each sample.

Authors’ reply:

We mentioned in “section-2.3. Particle diameter and surface charge” that the measurements were performed  in triplicate at room temperature and at 90° as scattering angel. Other details (Like stabilization time , run, …etc) were set in the device and the sample are routinely measured under the predetrmined conditions. Although we think that they are not so much effective (as long as they fall in well established range), such details were included in the manuscript. All changes have been yellow highlighted in the manuscript.

  • Section 2.4. What is the composition of the 0.22 micron filter used to filter the supernatant? Filter composition may condition the amount of drug entrapped during the filtration process.

Authors’ reply:

We used 0.20 micron filter (you can find below our own-captured image of the used type) during quantification of QC by HPLC. We have corrected the filter pores in the manuscript (becomes 0.20) as it was typing error. According to manufacturing instruction, the membrane is composed of hydrophilic polyester and frequently used for HPLC purposes (Suitable for aqueous, polar, hydrophilic as well as nonpolar, organic, hydrophobic solvents/samples with low particle-load). More details about the used filter are present at the link below.

https://www.mn-net.com/syringe-filters-labeled-chromafil-xtra-pet-13-mm-0.2-m-729222

  • Section 2.4. Has the HPLC method been previously validated? If so, it would be advisable to include information about linear behaviour (correlation coefficient), concentration interval, accuracy, LOQ and LOD, among others.

Authors’ reply:

Of course the method was validated. As we see that the inclusion of such details was not the scope of our work, preferred to add in supplementary materials. The details be have been added to supplementary materials.

Optimization of UHPLC conditions

The effective determination of QC was achieved on ACCLAIM™ 120 C18 column and isocratic elution at flow rate of 0.7 mL/min with a mobile phase consisting of water acidified by phosphoric acid (pH ~3) and acetonitrile. A different proportion of water acidified by phosphoric acid (pH ~3) and acetonitrile was tested; (40: 60 v/v). Also the detection wavelength and column temperature were tested and 210 nm and 25 °C were chosen. Under the optimum condition; the total run time of the HPLC method was 7 min. QC was detected at 5.27 min as retention time as shown in Fig. S1. Calibration curve was linear over a range of 0.02-80 µg/mL (Fig. S2), and the equation was A=1.3633C +0.2755 with a correlation coefficient (r2) value of 0.9998 as depicted in Table S1.

Fig. S1. HPLC chromatogram of QC

Figure S2: Calibration curve of QC

Table S1. Results of assay validation parameters of the proposed HPLC method for determination of QC

Parameter

QC

Range

0.02- 80 µg/ mL

Slope

1.3633

Intercept

0.2755

R2

0.9998

LOD*

0.004 µg/ mL

LOQ*

0.0133 µg/ mL

Recovery %

94.41- 98.19

RSD%

0.66 - 2.80

  • Section 2.6. What was the sample preparation process for this study? Were the nanoparticles kept in suspension during the study or kept lyophilized and reconstituted at the time of measurement? If lyophilized, did the authors employ any type of cryoprotectant for the freeze-drying of nanoparticles? The authors should give some insights about this observation.

Authors’ reply:

Stability study was carried out to the optimized formulation as nanodispersion without lyophilisation. We have added this in the section. All changes have been yellow highlighted in the manuscript.

  • Section 2.6. It is stated that the F0 formulation is prepared either without olive oil or without Pluronic F127, so it is understood that two different types of F0 formulation (F0a and F0b) are used. If so, in which assays is one used and in which the other?

Authors’ reply:

No, only one formulation (as reference formulation) was used; F0. It is a conventional phytosomal complex without adding surfactant or olive oil. We have rephrased the statement to be more clear.

  • Section 2.7. Why is PBS/Ethyl Alcohol 70:30 solution used, instead of using PBS exclusively?

Authors’ reply:

In order to maintain the sink conditions of QC. In our previous work (Elmowafy wt al., Polymers, 2021, 31;13(11):1808. doi: 10.3390/polym13111808.), we found that the solubility of QC in PBS was 0.85 mg/ml which was not sufficient to maintain the sink conditions. So, we added ethyl alcohol to increase the solubility of QC. As we used the same drug batch, we used the same conditions during in vitro release study. All changes have been yellow highlighted in the manuscript.

  • Section 2.7. What type of diffusion cells were used? Horizontal or vertical Franz diffusion cells?

Authors’ reply:

Vertical type was used. All changes have been yellow highlighted in the manuscript.

  • Section 2.8. Why have the size, PDI and Z-potential of HA-coated NPs not been evaluated if they are really intended to be the final formulation? This information should be included in the new version of the manuscript.

Authors’ reply:

  1. Required evaluations have been added in new section “3.5. Evaluation of HA based QC phytosomal complex.”

  • Section 2.9. Why was this concentration of quercetin taken as the study concentration, and is there any evidence on which this choice was based?

Authors’ reply:

Suitable reference has been added.

  • Section 2.9. Tween 80 was used in the ex vivo transcorneal permeation study but was not included in the materials section. What is the rationale for using that concentration of Tween in the donor portion along with the buffer solution?

Authors’ reply:

  1. It has been included in Materials section It was added in receprot compartment to improve the solubility of QC in PBS.

  • Section 2.9. Why are different volumes used in the donor and receptor chambers of this assay with respect to the in vitro assay. Are different Franz cell types used?

Authors’ reply:

Only one type of Franz cell was used. There were typing error in “section-2.9. Ex vivo transcorneal permeation”. We have corrected them. Receptor compartment was 12 ml and donor was equivalent to 20 mg of QC (2 ml of the formulation).

  • Section 2.10. No post hoc analysis was applied in the different studies. Multiple comparisons test (Tukey or Bonferroni test, among others) should be applied among formulations to deeply analysis the preclinical data.

Authors’ reply:

Means of the results were compared using Tukey’s multiple comparison testing using GraphPad Prism v.5. We have added in the relevant section. All changes have been yellow highlighted in the manuscript.

  • Section 3.2. PDI is not considered. PDI data should be included along with the other data, since it provides information on the homogeneity of the nanoparticle populations and, subsequently, about the reproducibility of the processing method.

Authors’ reply:

Section 3.2. was used to discuss interaction possibility between the components (FTIR). However, we considered PDI as a factor during stability study of the optimized formulation which verified the homogeneity of the optimized formulation. We also added PDI measurement of the optimized formulation after incorporation of HA.

  • Section 3.1.4. The units of measurement of the ratios and surfactant concentration were not indicated.

Authors’ reply:

It has been mentioned in section-“2.2. Fabrication of olive oil/phytosomal complexes” as three concentrations (0.5%, 1% and 2% w/v) 

  • Section 3.3. Has the reason for this phenomenon of size increase of the F0 formulation been studied? The Z potential values seem to be conserved independently of time and temperature, so the repulsion phenomenon between particles should be preserved. The authors should give some insights about this observation.

Authors’ reply:

Actually F0 was considered as control formulation. So, we focused in our study on the optimized formulation and discuss the effect of addition of olive oil and/or Pluronic on the behaviour of the formulation. So, the presence of Pluronic (polymeric surfactant) resulted in higher stability due to its steric effect.

  • Section 3.4. Why was the in vitro release study carried out for only 8h? Since these are controlled-release dosage forms, it would be ideal to evaluate the release profile over longer time periods to assess the possibility of reducing the frequency of administration, while preserving efficacy, and thus increase patient adherence to treatment. The controlled release performance of nanoparticles is related to their dispersion state, particle size and morphology.  How to ensure stable controlled release performance when these three parameters change?

Authors’ reply:

Performing in vitro release for 8 h was sufficient to release about 69.3% in case of optimized formulation, and 51.6% from F0. Within first 2h, around 37.4% and 30.5% was released from optimized formulation and F0 respectively. Dispersion state, particle size and morphology were not changed.

  • Section 3.4. The F0 formulation generally shows a longer half-life, in some cases almost double that of the optimized formulation. Should this be considered a disadvantage?

Authors’ reply:

As we mentioned above, F0 was considered as control formulation. So, we should make a balance between in vitro release/controlled release pattern and corneal permeation. However, in our future study (Part-II) we will investigate the in vivo pattern of both formulations and conclude the suitability of both formulations for clinical application.

  • Section 3.4. A quercetin suspension was not included in the release study as a second control formulation. The authors should give some insights about this observation.

 Authors’ reply:

F0 was considered as control formulation in that experiment and we preferred to study QC dispersion in ex vivo study.

Tables and figures:

  • Table 2. It would be appropriate to separate the data related to the nanoparticles composition and the results obtained after the systems elaboration in two different tables. Each table should be placed in the corresponding section (methodology and results, respectively).

Authors’ reply:

  1. It has been separated.

  • Table 2. The units of measurement of the ratios and surfactant concentration were not shown.

Authors’ reply:

It has been mentioned in section-“2.2. Fabrication of olive oil/phytosomal complexes” as three concentrations (0.5%, 1% and 2% w/v). It has been also added in the table. 

  • Table 5. The data in this table should be graphically represented to improve the reader's understanding of the data.

Authors’ reply:

We think that the data was presented in suitable and balanced pattern of tables and figures.

Minor changes:

  • There are formatting errors in the organization of the results section (sections 3.1.1 and successive sections after section 3.2).

Authors’ reply:

  1. It have been corrected.

Reviewer 2 Report

The manuscript prepared and evaluated quercetin-loaded hyaluronate based olive oil/phytosomal nanostructured carriers. The study approach is well-rationalized, and the conducted experiments are comprehensive across in vitro and ex vivo evaluations. However, the following issues should be addressed:

1.      The type of the provided manuscript should be modified from “review” to “original article”.

2.      Chemical structure of quercetin, as well as main components of the adopted nanocarriers should be presented.

3.      The authors performed dynamic light scattering (DLS) particle size analysis and measurement, Can the DLS graphs be incorporated in the manuscript or supporting information? Likewise, if possible and available, incorporate zeta potential graphs obtained from the instrument in the manuscript or supplementary data.

4.      Description of the animals used at the ex vivo trans-corneal permeation is missing, including sex and age.

5.      Legends for figures should be cited following the graphs/diagrams rather than before them.

6.      In Figure 4, it is recommended to label on the figure only the specific peaks corresponding to the characteristic structural functionalities of each component.

Additionally, authors should annotate for the missed OH peak within the quercetin spectrum as the possible interaction binding with the formulation components

7.      Table 5; Please provide the data as bar chart.

8.      In Figure 5, release data for sole quercetin should be presented.

9.      Author contributions are missing.

Author Response

The manuscript prepared and evaluated quercetin-loaded hyaluronate based olive oil/phytosomal nanostructured carriers. The study approach is well-rationalized, and the conducted experiments are comprehensive across in vitro and ex vivo evaluations. However, the following issues should be addressed:

  1. The type of the provided manuscript should be modified from “review” to “original article”.

Authors’ reply:

  1. It has been changed.
  2. Chemical structure of quercetin, as well as main components of the adopted nanocarriers should be presented.

Authors’ reply:

  1. It has been added in “Introduction-section”.
  2. The authors performed dynamic light scattering (DLS) particle size analysis and measurement, Can the DLS graphs be incorporated in the manuscript or supporting information?Likewise, if possible and available, incorporate zeta potential graphs obtained from the instrument in the manuscript or supplementary data.

Authors’ reply:

We have not mentioned particle size distribution graphs in the manuscripts as the manuscript will be overcrowded with graphs if we add additional 17 figures. However, the figures are listed below:

  1. Description of the animals used at the ex vivotrans-corneal permeation is missing, including sex and age.

Authors’ reply:

  1. Description of the animals has been added.
  2. Legends for figures should be cited following the graphs/diagrams rather than before them.

Authors’ reply:

  1. They have been corrected.
  2. In Figure 4, it is recommended to label on the figure only the specific peaks corresponding to the characteristic structural functionalities of each component.

Additionally, authors should annotate for the missed OH peak within the quercetin spectrum as the possible interaction binding with the formulation components

Authors’ reply:

  1. Characteristic peaks have highlighted by adding vertical dashed lines with function groups.
  2. Table 5; Please provide the data as bar chart.

Authors’ reply:

Data of release was fitted with different kinetic models. The data was entered in the software as mean values of percentages of drug release at different time intervals. So, no SD values can be obtained.

  1. In Figure 5, release data for sole quercetin should be presented.

Authors’ reply:

F0 was considered as control formulation in that experiment and we preferred to study QC dispersion in ex vivo study.

  1. Author contributions are missing.

Authors’ reply:

Ok. Author contributions have been added.

Round 2

Reviewer 1 Report

The aim of the work is very interesting, and part of the studies are adequate but contrary to the title, the article does not place enough emphasis on the formula optimization and evaluation processes, founding different inaccuracies in several places that hamper the interpretation of the methods and the assessment of the results.

It should be noted that basic qualitative and quantitative characterization studies of a nanometer-sized extended-release dosage form are not included in the manuscript. Despite the reviewer's suggestions to include such studies, the authors merely state that they will be studied in a " Part II of the manuscript". Nevertheless, the reviewer considers it important to emphasize that without a solid preclinical and evidence base, it would not be useful to proceed to in vivo studies. Studies as fundamental as morphological analysis by SEM or TEM analysis, as well as simple quantitative analysis including encapsulation efficiency, loading capacity or production yield should be a primary focus before moving on to more complex studies.

As mentioned in the first review, the paper could be potentially helpful to the scientific community but is not currently at that state of completion. One large issue was that the methods section had poorly information that would make these studies reproducible, and without that critical information, the reviewer also could not deeply interpret if these studies were appropriately performed. In the revised manuscript, the authors advocate their position in some studies showing an approach to in vivo studies. However, their argument lacks reproducibility, since they have not described the methodology and design of the study and it has been carried out in a non-representative sample size, and may contain many biases that cannot be accurately evaluated. These types of arguments make the reviewer doubt the validity of the design and development of the experiments, as they do not consider basic methodological aspects.

This manuscript reads still immature even though it has been revised twice. Many of the weak points of the manuscript, previously mentioned in the first review, remained unchanged in the revised manuscript. Besides, this manuscript also suffers from some other inadequate experimental design and poor editing of the manuscript. In addition, some questions in the rebuttal letter seem to be not properly answered. It seems that the authors have not obtained or analyzed the data from the studies exhaustively or, failing that, have not known how to interpret the results correctly. In addition, the reviewer has the impression that the authors have deviated the answer to certain questions, giving answers that did not correlate with the previously formulated questions.

Thus, in the view of the reviewer, this manuscript may suit some other pharmaceutical journals focusing on formulation development than Pharmaceutics after improvement.
